# Optimization of a direct detection UV wind lidar architecture for 3D wind reconstruction at high altitude

Thibault Boulant[1], Tomline Michel[1], and Matthieu Valla[1]

[1]DOTA, ONERA, Université Paris Saclay, F-91123 Palaiseau - France

**Correspondence:** Boulant (thibault.boulant@onera.fr)

**Abstract.**

An architecture for a UV wind lidar dedicated to measuring vertical and lateral wind in front of an aircraft for gust load alleviation is presented. To optimize performance and robustness, it includes a fiber laser architecture and a Quadri Mach-Zehnder (QMZ) interferometer with a robust design to spectrally analyze the backscattered light. Different lidar parameters have been selected to minimize the standard deviation of wind speed measurement projected onto the laser axis, calculated through end-to-end simulations of the instrument. The optimization involves selecting an emission/reception telescope to maximize the amount of collected photons backscattered between $100\,\text{m}$ and $300\,\text{m}$, a background filter to reduce noise from the scene, and photo-multiplier tubes (PMT) to minimize detection noise. Simulations were performed to evaluate lidar performance as a function of laser parameters. This study led to the selection of three laser architectures: a commercial solid-state laser, a design of a fiber laser, and a hybrid fiber laser resulting in standard deviations on projected wind speed of $0.17\,\text{m}\,\text{s}^{-1}$, $0.16\,\text{m}\,\text{s}^{-1}$, and $0.09\,\text{m}\,\text{s}^{-1}$, respectively, at $10\,\text{km}$ of altitude. To reconstruct the vertical and lateral wind on the flight path, the lidar is addressed to four different directions to measure four different projections of the wind. We calculate analytically (and validate through simulations) the addressing angle with respect to the flight direction that minimizes the root mean squared error (RMSE) between the reconstructed vertical and lateral wind components and the actual ones, assuming turbulence that follows the Von Karman turbulence model. We found that the optimum angle for an estimation at $100\,\text{m}$ is about $50°$, resulting in an improvement of about $50\,\%$ compared to an angle of $15°$-$30°$ typically used in current studies.

## 1 Introduction

Altitude air flow velocity measurements with atmospheric lidar have various applications, including weather forecasting (Baker et al., 1995, 2014; Bruneau and Pelon, 2021; Witschas et al., 2022), determining true air speed from aircraft (Augere et al., 2016), analyzing wind fields around High Altitude Platforms (Karabulut Kurt et al., 2021), and turbulence detection for Gust Load Alleviation (GLA) (Regan and Jutte, 2012; Fournier et al., 2021). GLA involves actively reducing the loads caused by air flow velocity on wings using actuators that modify the aerodynamic profile of the aircraft based on the direction and strength of the encountered wind. While this method is not novel and has been employed previously with detectors measuring turbulence near the aircraft structure, the use of lidar allows for measuring the wind structure in advance (referred to as feed-forward GLA), providing time for actuators to respond to the turbulence encountered. This approach has the potential to significantly

enhance the performance of such a system. Implementing this method requires measuring the variation of vertical and lateral wind velocity typically $100\,\text{m}$ - $200\,\text{m}$ ahead of the aircraft (In the case of the Airbus XRF1 (Fournier et al., 2021), the optimal distance ahead of the aircraft is $91\,\text{m}$, giving the control system enough time to react). Feed-forward GLA helps reduce constraints on wing resistance during the aircraft design phase, enabling the use of longer wings or reducing wing weight to decrease aircraft fuel consumption. In addition, it will limit aircraft vibrations, particularly the effects of air pockets that can hurt passengers (Kaplan et al., 2005). For this measurement, a direct-detection UV lidar optimized for molecular scattering is the optimal choice, as the presence of molecules is guaranteed at all altitudes, and the GLA system is intended to operate throughout the entire flight. In such lidar systems, a laser beam is directed into the atmosphere, and the wind velocity projected onto the laser propagation axis is determined by analyzing the frequency shift of the backscattered light induced by molecule velocity (Doppler effect). This shift is measured using a spectral analyzer. The use of UV wavelengths maximizes the molecular signal, as Rayleigh scattering is proportional to $1/\lambda^4$, where $\lambda$ is the laser wavelength. To spatially resolve the measurement, laser pulses are employed, so that the signal at time $t$ (with pulse emission at $t = 0\,\text{s}$) corresponds to a signal reflected at range $z = ct/2$ , where $c$ is the speed of light. To determine the 3D wind, the laser must be directed along multiple angles relative to the flight path to reconstruct the vertical and lateral wind components.

The spectral analyzer is a critical component for lidar performance. A first method involves measuring two signals corresponding to the backscattered light passing through two narrow bandwidth filters positioned on each side of the measured Rayleigh spectrum. Changes in the spectrum position due to molecule velocities alter the intensity ratio of the two signals (approximately linearly), allowing for the retrieval of the projected velocity (Garnier and Chanin, 1992). The primary limitation of this analyzer is that atmospheric temperature, pressure, and the presence of particles can alter the spectrum's shape, introducing biases during wind speed reconstruction. A second method involves interfering the backscattered light with itself by introducing a delay (induced by an optical path difference - OPD) between the two paths of the interferometer. In this case, the interference intensity is used to determine the phase difference between the two beams, which depends linearly on the frequency of the backscattered light. However, this intensity depends on several parameters: $I_{\text{OPD}} = A[1 + M \cos(\Delta\varphi_{\text{OPD}})]$ where $A = TI_0$, $T$ is the global transmission of the interferometer (i.e., the multiplication and addition of transmissions and reflections of the optics), $I_0$ is the intensity of the input light, $M$ is the contrast of the interference when varying OPD, $\Delta\varphi_{\text{OPD}} = \Delta\varphi_{0,\text{OPD}} + \delta\varphi_{\text{OPD}}$ is the phase difference between the two beams, $\Delta\varphi_{0,\text{OPD}}$ is the phase difference for the laser frequency, and $\delta\varphi_{0,\text{OPD}}$ is the phase difference induced by the molecule velocity. The phase $\Delta\varphi_{0,\text{OPD}}$ is determined by sending a sample of the laser pulse into the interferometer. To deduce the three other parameters ($A$, $M$ and $\delta\varphi_{\text{OPD}}$), measurements are performed for several OPDs separated by less than a wavelength to determine the interference oscillation pattern [i.e., $\cos(\Delta\varphi_{\text{OPD}})$]. For this method, different instruments have been developed, including a Mach-Zehnder (MZ), a Quadri Mach-Zehnder (QMZ) (Bruneau, 2001), a fringe imaging Michelson (FIM) (Cézard et al., 2009; Herbst and Vrancken, 2016), and a fringe imaging Mach-Zehnder interferometer (FIMZ) (Bruneau, 2002). The MZ provides the lower error on the wind speed but only gives 2 measurements for three parameters ($A$, $M$ and $\delta\varphi$), so one parameter (typically $M$) needs to be determined independently, and the error made on this parameter introduces biases in the wind speed measurement. Additionally, to minimize errors with the MZ, the wavelength of the laser and the OPD of the interferometer need to fulfill $\text{OPD} = m\lambda_0$ (with $m$ an integer) (Bruneau and Pelon,

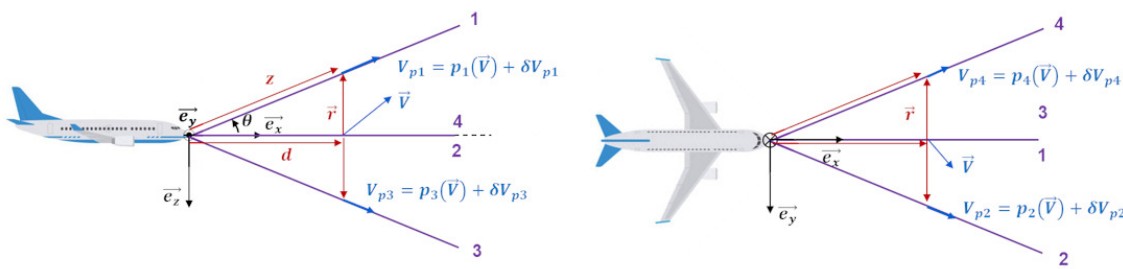

**Figure 1.** Measurement geometry of the reconstruction with the linear least square method with four axis. The lidar is located in the nose of the plane. $d$ is the range between the lidar and the estimation point on the flight path, $z$ the range of the point of the projections from the lidar, $\boldsymbol{r}$ the displacement vector between the projections point and the estimation point.

2021), requiring additional systems to lock the laser wavelength at the intersection of the two transmission curves. The QMZ, the FIM, and the FIMZ interferometers measure more than three values of OPD, allowing the retrieval of the three parameters without any assumptions (at the cost of a factor $\sqrt{2}$ increase in statistical error due to the desensitization of the interferometer to the backscattering ratio (Bruneau, 2001, 2002), , and do not require laser stabilization). Moreover, atmospheric parameters such as temperature, pressure, and the backscatter ratio do not produce bias. In the case of the FIM and FIMZ, the fringes need to be imaged by a set of detectors. Consequently, some signal is lost between cells of the imager, and the detectors are more expensive. On the other hand, the QMZ only needs four detectors, and each detector measures the entire signal at each output.

The second critical element is the UV laser system, which can be either diode-pumped and injection-seeded solid-state lasers (Lux et al., 2020) or microchip lasers amplified in free space (Wirth et al., 2009). This type of laser has the disadvantage of being sensitive to vibrations due to the number of free space optics, particularly in the case of ramp and fire lasers where there are presence of piezo-monitored optics to maintain the laser cavity adapted to the injected wavelength. These challenges have led to significant developments for DELICAT Laser (Vrancken et al., 2016) (similar to WALES (Wirth et al., 2009)) and AEOLUS laser (even though, in this case, the difficulty also involved the space environment (Mondin et al., 2017; Lux et al., 2021)). In this regard, fiber laser systems can lead to better performances for on-board direct detection wind lidar. Additionally, the oscillator part of fiber lasers and their fiber amplification stage have the advantage of being robust to vibrations, lighter, and more cost-effective when commercialized compared to the analogous part of solid-state lasers. Moreover, fiber lasers have the advantage to offer a better control of the pulse parameters (duration, cadence, timing). However, the free-space part, which includes the frequency tripling stage and any free-space amplifier, is still just as sensitive to vibration as solid-state lasers. Advances in fiber harmonic generation may therefore lead to an all-fiber laser insensitive to vibration.

The third critical aspect concerns the method applied to measure the vertical and lateral wind components. To reconstruct the 3D wind, a typical method consists in directing the lidar in four directions, making an angle $\theta$ with the central axis where the wind is reconstructed. This method was used for laser anemometer (Kliebisch et al., 2023) or to perform GLA (Rabadan et al., 2010; Kikuchi et al., 2020). For GLA, the lidar can be pointed upward and downward to measure the vertical component and to the left and right to measure the lateral component (see Fig. 1). For the different cases, the angle $\theta$ is generally chosen between

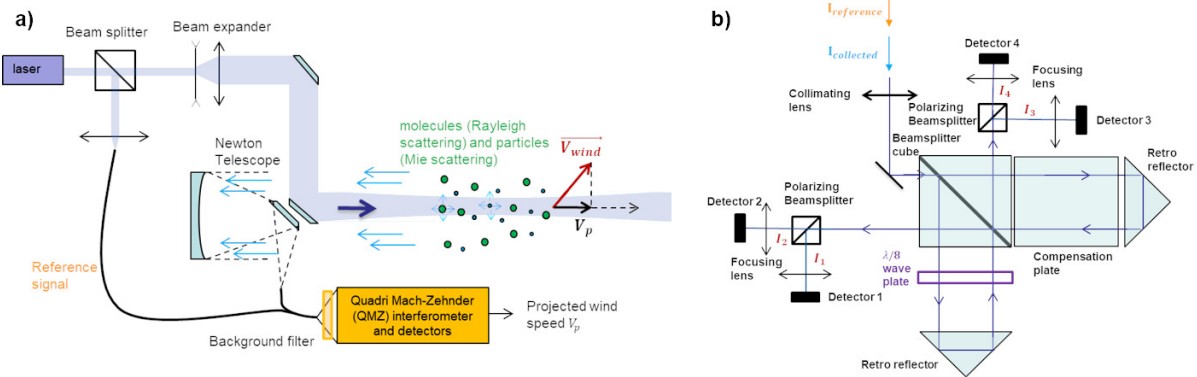

**Figure 2.** a) Schematic of the monostatic architecture with separated emission/reception optics chosen for the UV Doppler Wind lidar. b) Schematic of the Quadri Mach-Zehnder

$15°$ and $30°$ to satisfy the condition of a quasi-homogeneous wind field, as the measurement points are close to each other and, therefore, more correlated. However, the error committed in the reconstruction of the two wind components for small angles is given by $\frac{f(\delta V_p)}{2\tan\theta}$, where $f(\delta V_p)$ is a function depending on the error $\delta V_p$ induced by turbulence on the projections. The factor $\frac{1}{2\tan\theta}$ can lead to significant error amplifications ($\approx 1.9 - 0.9$ for $\theta = 15° - 30°$). To our knowledge, none of the studies have optimized the angle to minimize the error in the reconstructed 3D wind in the case of turbulence.

In this article, we present a study in which we have optimized the architecture of a molecular lidar designed to measure the lateral and vertical wind in front of an aircraft for GLA applications, aiming to maximize its performance by minimizing the error in the reconstructed wind. The design includes a robust Quadri Mach-Zehnder (QMZ) interferometer, fiber laser architectures, and an optimization of the lidar angle.To optimize the lidar architecture, an end-to-end simulator was developed to determine the collected light, the signal-to-noise ratio (SNR) on the detectors, and the calculation of the error in the wind

measurement projected on the lidar axis for a QMZ spectral analyzer. This simulator was used to optimize the telescope architecture, the detectors, the solar filter, and the laser parameters. Specifically, we determined lidar performance (see Table A1 and B1 for the parameters used) based on laser parameters and derived designs for a solid-state laser, a fiber laser, and a fiber laser followed by free-space amplifiers (hybrid fiber laser). For the vertical and lateral wind reconstruction, we optimized a design where the lidar is directed along four directions (up, down, left, and right) to measure the three components of the

wind vector. This involves minimizing the analytical calculations of the error in the estimation of the vertical and lateral wind components with the lidar angle. In the case of turbulence described by the Von Karman model, the error is minimized for an angle of about $50°$. This improves the root mean suqared error (RMSE) by about $50\%$ compared to the typical design using an angle of $15°$-$30°$. These results were confirmed by simulations of an aircraft traveling over $8\,\mathrm{km}$ through turbulence described by the Von Karman model.

## 2 Lidar architecture optimization

First, we present the architecture of the lidar and the robustified QMZ interferometer. Secondly, we present the simulator that was used for the optimization of the lidar. Thirdly, we utilize the simulator to optimize various components of the lidar, including the emission/reception telescope, the solar filter, the detectors, and the laser parameters.

### 2.1 Lidar architecture

The architecture of the direct detection UV Doppler wind lidar is shown in Fig. 2.a). The laser emits pulses at $355\,\text{nm}$ for solid-state lasers and at $343\,\text{nm}$ for fiber lasers. A beam splitter is inserted to take a sample of the laser, serving as a reference signal. The laser light passes through a beam expander that focuses the laser at long distances. A Newton telescope is used to collect the signal backscattered by molecules and particles and to focus it into a multimode fiber with a numerical aperture of 0.22. The signal goes through a solar filter to greatly reduce the light coming from the background. A fiber coupler is used to combine the collected backscattered light and the reference signal. The long fiber is employed either on the reference signal or on the backscattered signal to temporally separate the two signals at the input of the interferometer. Both signals pass through a spectral analyzer. The intensities measured by the detectors are compared for the two signals to determine $\delta\varphi$ and to deduce the projected wind speed. It should be noted that the reference signal also allows obtaining absolute synchronization of the measured signal.

The QMZ interferometer is shown in Fig. 2.b). The signal, coming from the multimode fiber, is collimated by a converging lens and passes through a 50/50 beam splitter cube, which splits the beam into two arms of different lengths. A $\lambda/8$ wave plate on the short arm increases the OPD for horizontal polarization by $\lambda/4$ considering the round trip, compared to vertical polarization. On the long arm, a glass plate is used to reduce beam divergence and improve the overlap of the two beams on the detector. This leads to an increase in the angular acceptance of the interferometer (Smith and Chu, 2016). The beams from both arms are combined on the 50/50 beam splitter cube. The OPD of the output that has undergone an odd number of reflections is shifted by $\lambda/2$ relative to the other. On each output, a polarizing beam splitter cube separates the horizontal and vertical polarizations. This gives four outputs with OPDs given by $D_0$, $D_0 + \lambda/4$, $D_0 + \lambda/2$ and $D_0 + 3\lambda/4$ for $I_1$, $I_2$, $I_3$ and $I_4$, respectively, where $D_0$ is the OPD for output $I_1$. These outputs are focused on detectors that convert optical signals into current. The current is converted to voltage, which is then sampled and digitized into a computer. Fig. 3 shows the evolution of the simulated signal at the output of the detectors, for the reference signal and the Rayleigh signal. Signal processing makes it possible to recover the phase of the interference, the frequency offset, and finally the projected wind speed.

We have chosen an architecture that includes one 50/50 beam splitter cube for the separation and recombination of the beams, along with two retroreflectors forming the two arms, to create a robust interferometer. Indeed, the QMZ is not sensitive to angular misalignment. However, simulations (Boulant et al.) show that the retroreflector relative to each other needs to be positioned within about $2\,\mu\text{m}$. This is achieved using an X,Y mount on one retroreflector.

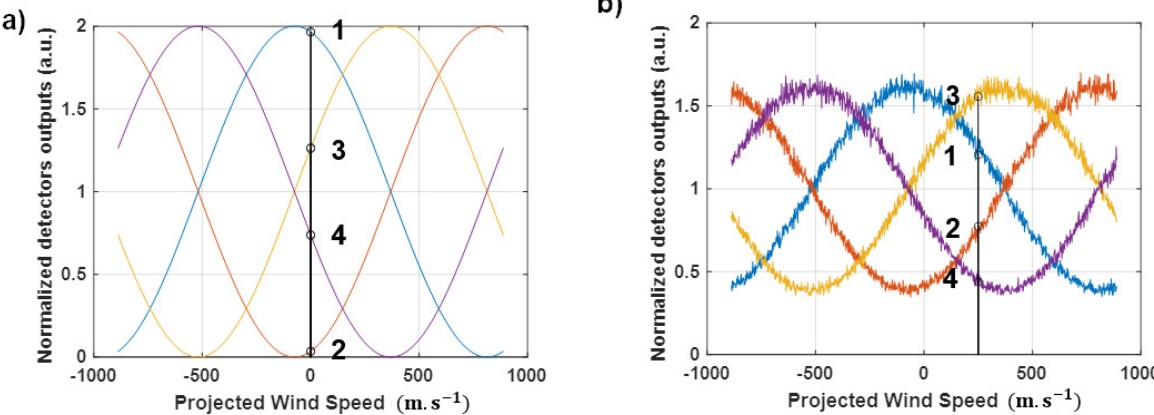

**Figure 3.** Signal at the detectors outputs for a) the reference signal and b) the Rayleigh signal. All signals are normalized by the mean value of the outputs. The black line highlight the values of the signal for the reference signal ($0\,\mathrm{m\,s^{-1}}$) and for the Rayleigh signal when the wind speed is $250\,\mathrm{m\,s^{-1}}$ (relative wind when plane fly at $250\,\mathrm{m\,s^{-1}}$ at $10\,\mathrm{km}$ altitude)

## 2.2 UV lidar simulator

A simulator that use analytical formula has been developed to optimize the lidar architecture for the GLA application. It comprises three steps. The first step calculates the emission/reception overlap function, that is to say the ratio between the light entering the fiber and the light collected by the pupil. The second step assesses the SNR at the outputs of the detectors, and the third step determines the standard deviation of the wind speed to evaluate the overall lidar performance.

To calculate the overlap function, we use the method presented in the thesis manuscript of [Cezard (2008) (part 3.2.)] and [Liméry (2018) (part 3.3.4)]. The simulator first computes the laser propagation using the Gaussian beam approximation. For each point along the laser beam, described by a distance $z$ from the telescope and $\rho$ from the laser axis (assuming cylindrical symmetry), the simulator determines the image of the point by the first pupil and the amount of light that enters the fiber that is $\frac{S(\rho,z)2\pi\rho d\rho}{S_{\mathrm{pup}}\pi\omega(z)^2}$. $\omega(z)$ is the radius of the laser beam at $1/e^2$, $S_{\mathrm{pup}}$ is the surface of the first pupil and $S(\rho,z)$ the surface of the pupil corresponding to the collected rays that goes to the fiber. Subsequently, for each distance from the telescope, the simulator integrates all the quantities obtained with all the points of the laser beam to calculate the ratio of the light that enters the fiber to the light collected by the pupil that is the overlap function $\gamma(z) = \frac{1}{S_{\mathrm{pup}}\pi\omega(z)^2}\int_0^{\omega(z)} S(\rho,z)2\pi\rho\,d\rho$. The configuration is optimized for a given distance when the overlap function is equal to one.

Secondly, the simulator calculates the total SNR, which is the collected light integrated over a range gate to the square root of the quadratic sum of all noise. To calculate the signal, we use the molecular backscattering and absorption coefficients determined with the evolution of the molecular density calculated using the US standard atmosphere model (Atmosphere, 1976). Particle backscattering and absorption are neglected at high altitudes because the concentration of particles is very low, leading to a backscattering and absorption much lower than the molecular backscattering and absorption (Vrancken et al.,

2016). Then, the previously determined overlap function is used to calculate the collected signal. The total noise corresponds to the quadratic sum of the shot noise of the backscattered signal, the background noise, the speckle noise of the backscattered signal, and the detection noise, which includes the dark noise of the detector and the electronic noise (Fujii and Fukuchi, 2005, p. 488, 574 and 695). The configuration is considered optimized for a given average laser power (to the first order, proportional to the laser volume, this will be refine considering the different laser technology) when the noise is limited by the shot noise

of the backscattered signal. Indeed, this noise can only be reduced by increasing the laser power. In this case, the SNR is proportionnal to the average power of the laser $\sqrt{P_{\mathrm{av}}}$.

The third step of the simulator is dedicated to calculating the standard deviation of the wind speed for the total SNR in order to assess the measurement performance. For simplicity, we utilized the analytical formula derived by Bruneau and Pelon (2003) and incorporated the contribution of speckle noise into this formula:

$$165 \quad \sigma_{v_p}^2 = \left(\frac{c\lambda_0}{4\pi D_0}\right)^2 \frac{2}{N_{\mathrm{acc}} M_{\mathrm{tot}}^2 \mathrm{SNR}_{\mathrm{photon}}^2}\left(1 + F_B M_{\mathrm{tot}}^2 \frac{\sin(2\varphi)^2}{2}\right) + \frac{c\lambda_0}{4\pi D_0} \frac{2}{N_{\mathrm{acc}} M_{\mathrm{tot}}^2 \mathrm{SNR}_{\mathrm{detection}}^2}\left(1 + M_{\mathrm{tot}}^2 \frac{\sin(2\varphi)^2}{2}\right) + \sigma_{v_p,\mathrm{speckle}}^2$$

$$(1)$$

$N_{\mathrm{acc}}$ is the number of accumulated lidar shot during the measurement time, $M_{\mathrm{tot}}$ is the total contrast of the interferences, $\mathrm{SNR}_{\mathrm{photon}}$ is the SNR considering only shot noises (from backscattering signal and background signal), $\mathrm{SNR}_{\mathrm{detection}}$ considering only the detector and electronic noises, $F_B = \frac{N_{\mathrm{bkg}} - N}{N_{\mathrm{bkg}} + N}$ with $N_{\mathrm{bkg}}$ the number of background photons and we note $\varphi = \Delta\varphi_{\mathrm{OPD}}$ for simplicity. The calculation details of $\sigma_{v_p,\mathrm{speckle}}^2$ and some parameters are outlined in Appendix A. We verified that this

analytical formula was a good estimator using Monte Carlo simulations. In these simulations, we stochastically generated the currents obtained at the detector outputs based on the statistics of the various noise sources. Subsequently, the four currents were convolved with the detector impulse response. The Maximum Likelihood Estimator (MLE) (see paper of Cézard et al. (2009) for principle) was then employed to retrieved the projected wind speed along the laser axis, and we assessed the error distribution across multiple simulations. Furthermore, we confirmed that the results obtained using the analytical formula closely matched Cramer Rao's lower bound (Cézard et al., 2009) as we obtain a relative difference of $2\%$.

## 2.3 Optimization of the lidar performances

### 2.3.1 Emission/reception architecture

Several simulations were conducted to determine the optimal focusing distances for both the telescope and the laser, aiming to maximize the overlap function between $100\,\mathrm{m}$ and $300\,\mathrm{m}$ for our optical architecture. The setup includes a telescope with

180 a diameter of $152.4\,\mathrm{mm}$, a primary mirror focal length of $609.6\,\mathrm{mm}$, a second mirror obstruction diameter of $38\,\mathrm{mm}$, and a fiber diameter of $400\,\mu\mathrm{m}$. These calculations were performed considering a laser beam with a size of $30\,\mathrm{mm}$ upon exiting the lidar, where $M^2$, define as the ratio of the beam divergence angle to the beam divergence angle of the perfect Gaussian beam at the same wavelength, is considered lower than 8, value obtain for the commercial laser Merion C by Lumibird. The optimized focusing distances were found to be 155 m for the telescope and $100\,\mathrm{m}$ for the laser, resulting in an overlap function $\gamma$ equal

to 1 across the entire range.

### 2.3.2 Detector

We compared the detection noise level in terms of the number of photons for the specified range gate across three types of detectors: a PIN photodiode amplified with a transimpedance, an avalanche photodiode (APD), and a PMT (Hamamatsu S5971; Hamamatsu S9075; Hamamatsu R10721-210). Indeed, the total noise induced by the backscattered shot noise must exceed the total detection noise. Taking the definition of the excess noise factor from (PMT Handbook), we have $F = \left(\frac{\text{SNR}_{\text{input}}}{\text{SNR}_{\text{output}}}\right)^2$ with $\text{SNR}_{\text{input}} = \frac{N}{\sigma_{\text{input}}}$ and $\text{SNR}_{\text{output}} = \frac{G\eta N}{\sigma_{\text{output}}}$. $N$ represents the sum of backscattered photons obtained on the four detectors, $G$ stands for the gain of the detector, and $\eta$ signifies the quantum efficiency. The noise variance induced by the backscattered shot noise at the output of the detectors will be $\sigma_{\text{output}}^2 = F(G\eta)^2 \sigma_{\text{input}}^2 = F(G\eta)^2 N$. This noise must exceed the detection noise, leading to the condition $N >> \frac{4\sigma_{\text{det}}^2}{F(G\eta)^2}$, where $\sigma_{\text{det}}^2$ denotes the detection noise of a detector expressed in the number of electrons calculated for a range gate of $25\,\text{m}$. In order to meet the condition, we take $N > 10 \frac{4\sigma_{\text{det}}^2}{F(G\eta)^2}$, with the right term corresponding to the equivalent number of photons produced by the detection noise. For the PIN, we found $1.1 \times 10^8$, for the APD $2.8 \times 10^6$, and for the PMT $2.9 \times 10^{-5}$. Only the PMT ensures a low level of detection noise compared to the shot noise level.

### 2.3.3 Solar filter

The spectrally thin solar filter blocks most of the background signal (broad spectrum) that goes to the spectral analyzer but transmits the Rayleigh signal. Ideally, the filter bandwidth should encompass the spectral width of the Rayleigh signal plus and minus the maximum Doppler frequency shift ($\approx 1\,\text{pm}$ at $355\,\text{nm}$ for a wind speed of $100\,\text{m\,s}^{-1}$). However, the smaller the desired filter bandwidth, the smaller the transmission and the more expensive the component.

To optimize the filter, we need to have a photon noise of the background signal much smaller than the photon noise of the backscattered signal. We chose for the threshold $\frac{N}{N_{\text{bkg}}} > 10$ where $N_{\text{bkg}}$ is the sum of background photons of the four detectors. For this, we need:

$$E_P > BR\Delta\lambda z^2 \tag{2}$$

where $B = \frac{10\pi \text{FOV}^2 R}{2c\beta T_{\text{atm}}^2 \gamma(z)}$, $R$ is the background radiance, $z$ the range to the telescope, $\beta$ the backscatter coefficient, $T_{\text{atm}}$ the atmospheric transmission, FOV is the telescope field of view and $\Delta\lambda$ is the filter bandwidth (Full Width at Half-Maximum). We can see that the optimized filter depends on the laser parameters.

As the minimum energy of the pulse increases with distance, in our case Eq. 1 must be fulfilled at $300\,\text{m}$ to be fulfilled over the whole range. We also assume for the study a background radiance taken equal to $0.3\,\text{W\,m}^{-2}\,\text{sr}^{-1}\,\text{nm}^{-1}$. For a filter bandwidth of $1\,\text{nm}$, this results in a minimum laser energy per pulse of $298\,\mu\text{J}$. This filter bandwidth is used for the rest of the simulation, as it about corresponds to the limit of the technology in term of filter thickness.

### 2.3.4 Laser optimization on the ground and at 10 km of altitude

To optimize the laser parameters using the simulator, simulations were performed by adjusting the average laser power and pulse repetition frequency to assess the error in the retrieved wind velocity. We neglected electrical noise by considering PMT

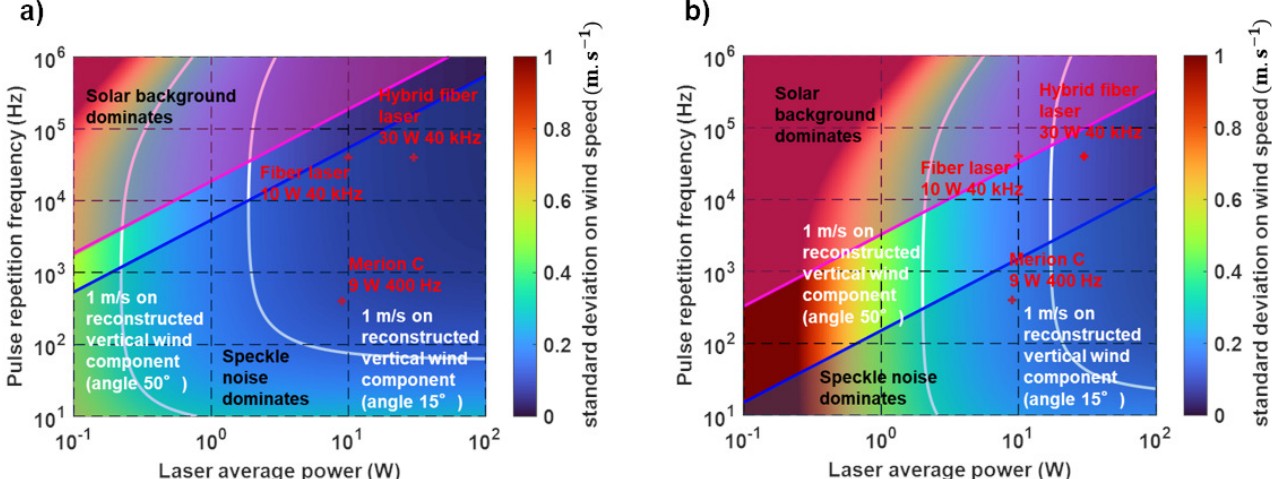

**Figure 4.** Standard deviation on wind speed estimation as a function of laser average power and pulse repetition frequency at $150\,\mathrm{m}$ from the lidar, considering molecular scattering and particles scattering, for a) wind measurement on the ground and b) at $10\,\mathrm{km}$ altitude. The white line shows the limit of $1\,\mathrm{m\,s^{-1}}$ on the reconstructed vertical wind component for a lidar angle $15°$ and $50°$.

detectors. At low altitudes (less than $1\,\mathrm{km}$), we assumed a backscatter coefficient for particles of $8 \times 10^{-6}\,\mathrm{m^{-1}\,sr^{-1}}$. The backscatter coefficient for molecules is $7.2 \times 10^{-6}\,\mathrm{m^{-1}\,sr^{-1}}$ on the ground and $2.1 \times 10^{-6}\,\mathrm{m^{-1}\,sr^{-1}}$ at $10\,\mathrm{km}$ of altitude. The simulations were performed at a distance of $150\,\mathrm{m}$ from the telescope on the laser axis, which corresponds to the intended 3D
wind reconstruction distance of about $100\,\mathrm{m}$ in front of the aircraft. Additionally, we assumed a range gate of $25\,\mathrm{m}$ to match the GLA specifications. The measurement times were set to $0.1\,\mathrm{s}$, corresponding to an integration over $25\,\mathrm{m}$ along the aircraft direction traveling at $250\,\mathrm{m\,s^{-1}}$. We considered that the laser has a full width at $1/e^2$ of $400\,\mathrm{MHz}$, significantly less than the spectral broadening induced by the thermal movement of the molecules ($6.3\,\mathrm{GHz}$ for a full width at $1/e^2$ (Bruneau and Pelon, 2003)). For the Mie scattering, the coherence time is limited by the laser pulse duration, i.e. $10\,\mathrm{ns}$. For the Rayleigh scattering,
it is limited by spectral broadening due to thermal motion of the molecule, i.e. $0.63\,\mathrm{ns}$ for a broadening of $6.3\,\mathrm{GHz}$. The simulations were conducted both on the ground and at $10\,\mathrm{km}$ altitude, approximately corresponding to the aircraft's cruising altitude, as the GLA must operate throughout the flight.

Fig. 4.a) shows the evolution of the standard deviation of the projected wind speed computed on the ground with the Eq. (1). The white lines give the laser parameters resulting in an error, equivalent to $3\sigma$ (where $\sigma$ denotes the standard deviation),
of $1\,\mathrm{m\,s^{-1}}$ on the reconstructed vertical wind component for a lidar angle $15°$ and $50°$. The methodology utilized for wind calculation is elaborated in section (3.1). This corresponds to a standard deviation on the projected wind speed of $0.12\,\mathrm{m\,s^{-1}}$ for $15°$ and $0.35\,\mathrm{m\,s^{-1}}$ for $50°$. The magenta line denotes the threshold where the variance of the backscattered signal exceeds ten times the background noise variance at $300\,\mathrm{m}$. Beyond this threshold, the laser pulse energy diminishes, leading to an increased number of background photons compared to backscattered photons. The blue line represents the boundary where
the variance of the backscattered signal surpasses ten times the speckle variance of the backscattered signal at $100\,\mathrm{m}$. Below

this threshold, averaging the measurements fails to adequately average speckle patterns. Within the region delineated by these lines, the measurement is constrained by the shot noise of the backscatter, and the lidar is considered optimized. Within this range, performance is directly proportional to the average laser power (laser parameter serving as an indicator of laser size).

Fig. 4.b) shows the evolution of the standard deviation of the wind speed computed at $10\,\mathrm{km}$ altitude, featuring the same threshold lines as in Fig. 4.a). Performance is diminished because of the absence of particles and the reduced density of molecules at this altitude, resulting in a decreased amount of backscattered signal. Additionally, the scarcity of signal amplifies the impact of the background signal, as evidenced by the magenta line being lower than that calculated on the ground. Conversely, the speckle noise decreases because the backscattering is predominantly molecular, which is less coherent than particulate backscattering. Moreover, achieving an accuracy of $1\,\mathrm{m\,s^{-1}}$ is more challenging at this altitude when taking a lidar angle of $15°$. However, as we will see in section 3., the lidar angle can be increased to $15°$ and for this angle the limit on the standard deviation is $0.35\,\mathrm{m\,s^{-1}}$, so all the three laser allow to reach the precision of $1\,\mathrm{m\,s^{-1}}$ on the vertical wind component.

### 2.3.5 Selected laser configurations

We have used the Fig. 4 to select laser parameters for three laser technologies: seeded solid states laser, fiber laser and hybrid fiber laser.

Regarding the seeded solid-state laser, the primary challenge lies in achieving a high repetition rate to avoid being constrained by speckle noise. We have opted for the injection-seeded Merion C commercialised by Lumibird, which boasts a repetition rate of $400\,\mathrm{Hz}$ and delivers $22.5\,\mathrm{mJ}$ of energy per pulse. In this scenario, a wide bandwidth solar filter can be employed (typically » $1\,\mathrm{nm}$ is chosen). This configuration yields a standard deviation $\sigma_{\mathrm{lidar}}$ of $0.11\,\mathrm{m\,s^{-1}}$ at low altitude and $0.17\,\mathrm{m\,s^{-1}}$ at high altitude.

The second technology is a fiber laser, made with doped and pumped fiber, emitting at $1\,\mathrm{\mu m}$, with a frequency tripling stage. Because of the limit in peak power due to the Brillouin effect in the fiber, we chose a high repetition rate of $40\,\mathrm{kHz}$ well adapted to the fiber laser. We estimated that the maximum average power of $10\,\mathrm{W}$ can be achieved with current technology, it allows the use of a solar filter up to $0.84\,\mathrm{nm}$. So with a filter of $1\,\mathrm{nm}$, the projected wind speed measurement at $300\,\mathrm{m}$ will then be slightly affected by background noise. The results for standard deviation $\sigma_{\mathrm{lidar}}$ are $0.05\,\mathrm{m\,s^{-1}}$ at low altitude and $0.16\,\mathrm{m\,s^{-1}}$ at high altitude.

The third configuration is the hybrid fiber laser, obtained by adding a free space amplifier at the output of the fiber laser. We estimated that the maximum average power of $30\,\mathrm{W}$ can be reached, which allows the use of a solar filter up to $2.5\,\mathrm{nm}$. This results in a standard deviation $\sigma_{\mathrm{lidar}}$ of $0.03\,\mathrm{m\,s^{-1}}$ at low altitude and $0.09\,\mathrm{m\,s^{-1}}$ at high altitude.

The main parameters that have been used in the simulation are sum up in Table A1 and B1. The parameters in red correspond to the one that were optimized with simulations.

## 3 Wind reconstruction

We minimized the total error in the vertical and lateral components of the wind reconstructed using the four-axis design in the presence of turbulence. These components significantly influence lift, emphasizing the importance of accurate estimation to avoid errors when attenuating the tubrulence effect in GLA applications. Firstly, we establish the expression of the instrumental error contribution to the total error and evaluate it for the three laser designs previously established. Secondly, we define the expression of the turbulence contribution to the total error for Von Karman turbulence. Thirdly, we determine the lidar angle that minimizes the total error in the vertical component.

### 3.1 Lidar angle optimization with analytical method

The first step involves establishing the contribution of instrumental error to the total error in the vertical and lateral wind components, which depends on the error in the projections used to estimate the vertical component and the lidar angle. Referring to Fig. 1, the vertical wind component is given by $v_z = \frac{V_3 - V_1}{2\sin(\theta)}$, where $V_3$ and $V_1$ are projections of the wind onto axes 3 and 1, respectively. The lateral component is reconstructed using the same formula, employing projections from axes 2 and 4, $V_2$ and $V_4$. These projections are affected by the lidar noise $\sigma_{\mathrm{lidar}}$, evaluated in section 2.3.5 for the three laser designs. If we assume that the wind is homogenous, the error is obtained from the instrumental noise given by $\sigma_{\mathrm{Vz,instrumental}}(d,\theta) = \frac{\sqrt{2}}{(2\sin(\theta))}\sigma_{\mathrm{lidar}}(d,\theta)$. The error in the lateral component is the same due to symmetry. This error depends on $d$, the range of estimation along the flight path, and $\theta$ since $\sigma_{\mathrm{lidar}}$ depends on the range $z$ from the lidar, where $z = d/\cos(\theta)$. Moreover, if we suppose that the lidar is optimized, which is the case for the hybrid fiber laser between $100\,\mathrm{m}$ and $300\,\mathrm{m}$, the predominant noise is the backscattered signal shot noise. In this scenario, $\sigma_{\mathrm{lidar}}$ is proportional to the measurement range $z$ along the lidar axis. Given that the standard deviation for the three laser configurations was calculated at $150\,\mathrm{m}$, the instrumental noise at range $d$ is $\sigma_{\mathrm{lidar}}(d,\theta) = \frac{\sigma_{\mathrm{lidar}}(z_0)}{z_0}\frac{d}{\cos(\theta)}$. Therefore, the instrumental error contribution to the total error in the vertical wind component is:

$$\sigma_{\mathrm{Vz,instrumental}}(d,\theta) = \frac{\sqrt{2}}{(2\sin(\theta))}\frac{\sigma_{\mathrm{lidar}}(z_0)}{z_0}\frac{d}{\cos(\theta)} \tag{3}$$

We assess the error in the reconstructed vertical and lateral wind components for the three selected laser configurations. Assuming an estimation distance of $d = 100\,\mathrm{m}$ and an angle of $15°$, typically used for the lidar angle, we obtain errors of $0.32\,\mathrm{m\,s^{-1}}$, $0.30\,\mathrm{m\,s^{-1}}$, and $0.17\,\mathrm{m\,s^{-1}}$ for the Merion C laser, the fiber laser, and the hybrid fiber laser, respectively. This result ensures that the $3\,\sigma$ error for all three designs remains below $1\,\mathrm{m\,s^{-1}}$.

The second step involves establishing the expression of the turbulence contribution to the total error for turbulence described by the Von Karman model. This model provides expressions for power spectral densities that best match measured turbulence data (Giez et al., 2021), particularly in the inertial subrange where the energy cascade from large eddies to smaller ones occurs. The error in the vertical wind component, corresponding to the RMSE between the reconstructed component and the actual

component, if we only consider the turbulence contribution, is:

$$\sigma_{\text{Vz,turbulence}}(d,\theta) = \sqrt{\langle (V_z - V_{z0})^2 \rangle}$$

$$= \sqrt{\frac{2}{(2\sin(\theta))^2}(\langle \delta V_{p1}^2 \rangle + \langle \delta V_{p3}^2 \rangle - 2\langle \delta V_{p1} \delta V_{p3} \rangle)} \qquad (4)$$

where $V_{z0}$ is the vertical component of the actual wind, $\delta V_{p1}$ and $\delta V_{p3}$ are the differences between the projections of the real wind and the measured projections, for axis 1 and 3 respectively (see Fig. 1). The symbols $\langle \rangle$ account for the ensemble averaging. For a Von Karman turbulence, which is homogeneous and isotropic (i.e. the statistics are independent of the coordinate rotations), the error is (calculation details in appendix B, using the formulas presented in (Wilson, 1998)):

$$\sigma_{\text{Vz,turbulence}}(d,\theta) = \sqrt{\frac{D_{NN}(2r)}{(2\tan(\theta))^2} + \frac{3B_{LL}(0) + B_{LL}(2r)}{2} - 2B_{LL}(r)} \qquad (5)$$

where $r = d\tan(\theta)$ is the distance between the point on the lidar axis and the point on the flight path, $B_{LL}$ the longitudinal correlation function of the turbulence (for the wind component longitudinal to the displacement vector $r$) and $D_{NN}$ the structure function for the lateral component of the wind (lateral to the displacement vector $r$). Combining equations (52),(99) and (101) of (Wilson, 1998), we obtain expressions for the correlation and structure functions:

$$B_{LL}(r) = \sigma_S^2 \frac{2}{\Gamma(1/3)} \left(\frac{r}{2l}\right)^{1/3} K_{1/3}\left(\frac{r}{l}\right) \qquad (6)$$

$$B_{NN}(r) = \sigma_S^2 \frac{2}{\Gamma(1/3)} \left(\frac{r}{2l}\right)^{1/3} \left[K_{1/3}\left(\frac{r}{l}\right) - \left(\frac{r}{2l}\right) K_{2/3}\left(\frac{r}{l}\right)\right] \qquad (7)$$

$$D_{NN}(r) = 2(B_{NN}(0) - B_{NN}(r)) \qquad (8)$$

where $\sigma_S$ the standard deviation of the wind amplitude in the turbulence, $l$ the turbulence length scale, $\Gamma$ is the gamma function, $K_n$ is a Bessel function of the first kind and $B_{NN}$ the lateral correlation function of the turbulence. The standard deviation of the wind amplitude is related to the spectrum energy $E_v(k)$ of the wind field with the equation $\int_0^\infty E_v(k)\,dk = \frac{3\sigma_S^2}{2}$ (Wilson, 1998), where $k$ represents the spatial frequency.

The optimization of the lidar angle was performed by minimizing the expression of the total error on the vertical wind component with respect to $\theta$. The expression for the total error is $\sigma_{Vz} = \sqrt{\sigma_{\text{Vz,instrumental}}(d,\theta)^2 + \sigma_{\text{Vz,turbulence}}(d,\theta)^2}$. This error was evaluated for a range of estimation of $d = 100\,\text{m}$ at an altitude of $10\,\text{km}$, using the Merion C laser, and a Von Karman turbulence with $l$ equal to $762\,\text{m}$ (2500 ft) and $\sigma_S$, the standard deviation of the wind amplitude, equal to $10\,\text{ms}^{-1}$. Fig. 5.b) illustrates the evolution of the total error, that is the RMSE between the reconstructed and actual lateral component along the flight path, as a function of the lidar angle $\theta$. The RMSE reaches a minimum for a lidar angle of $51°$. The RMSE obtained for this angle is $7.2\,\text{ms}^{-1}$, nearly twice as low as the RMSE of $12.7\,\text{ms}^{-1}$ obtained for an angle of $15°$. Choosing an angle of $51°$ may seem counter-intuitive in case of turbulent wind, indeed the measurement points are much further apart for such an angle ($2r = 240\,\text{m}$ between the two opposite points for $50°$, while $2r = 54\,\text{m}$ for $15°$). Due to this larger distance, one might assume that the measurement points are not correlated, leading to significant errors in the projections, which affect the reconstructed wind component. In fact, for the wind structure we are examining (in the inertial subrange), in a turbulent wind

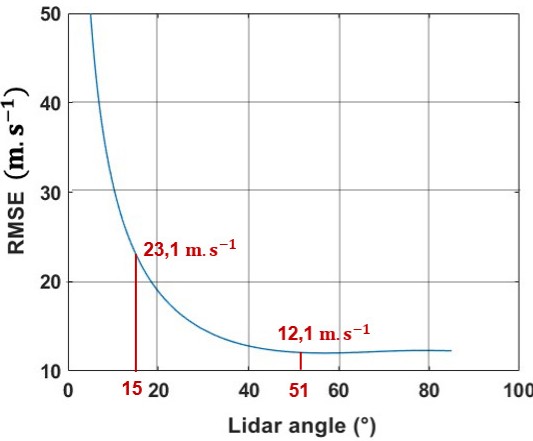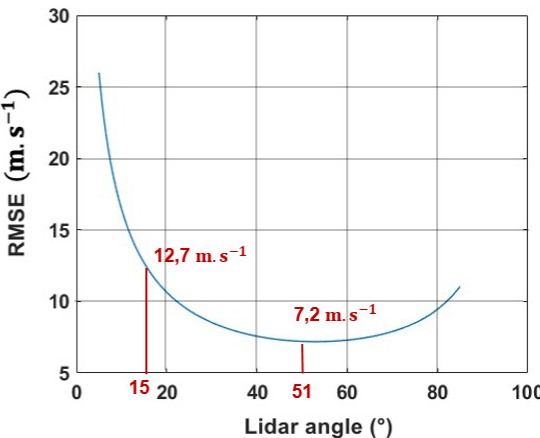

**Figure 5.** RMSE between the reconstructed and actual lateral component on the flight path as a function of the lidar angle from the flight direction $\theta$, for wind measurement at (a) low and (b) high altitude

field like Von Karman turbulence, large eddies are much more influential than small ones. The smaller wind structures induce only a minor error on all four projections, which can be amplified for small lidar angles due to the factor $1/(2\sin(\theta))^2$ in Eq. (3) when retrieving the wind component. For larger angles (above $50°$), the instrumental noise becomes significant because the measurement range along the lidar axis increases.

### 3.2 Optimized angle for measurements at low altitude

We study the evolution of the optimized angle for measurement on the ground. The measurement parameters remain the same: the vertical wind component is estimated at $d = 100\,\mathrm{m}$ from the lidar using the Merion C laser. The length scale of the Von Karman turbulence at low altitude is much lower and is assumed to be equal to $100\,\mathrm{m}$ for this study. Additionally, we assume a turbulence strength $\sigma_S$ of $10\,\mathrm{m}^2\,\mathrm{s}^{-2}$. The evolution of the RMSE at this altitude is depicted in Fig. 5.a) as a function of the lidar angle. We observe that the optimized angle is around $51°$. However, the error is higher than at high altitude, partly due to the fact that smaller eddies have more energy. This increases the difference between the projections of the real wind and the measured projections, thereby increasing the error.

### 3.3 Validation with simulations of turbulence at 10 km of altitude

To illustrate the improvement at $51°$, reconstruction simulations of the lateral wind component with two lidar angles were conducted. The simulation utilized a simulator that calculates the 3D wind field by following the statistics of the Von Karman turbulence model, using the equations presented in the Army Research Laboratory technical report (Wilson, 1998). The 3D spectra are calculated as a function of the spatial frequency: $\Phi_{ij}(k) = \frac{E_v(k)}{4\pi k^4}(\delta_{ij}k^2 - k_i k_j)$ where the indices $i$ and $j$ represent the direction ($x$, $y$, or $z$) and $\delta_{ij}$ is the Kronecker delta. For a Von Karman turbulence, $E_v(k) = 1.4528\frac{\sigma_S^2 k^4 l^5}{(1+k^2 l^2)^{17/6}}$. For each $k$,

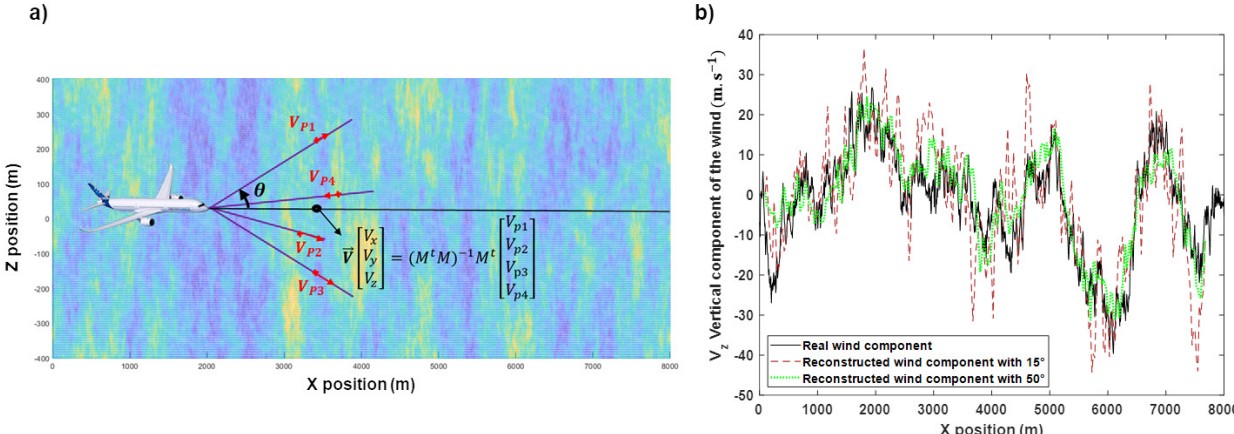

**Figure 6.** a) Sample of a wind simulation (norm of the wind vector, in the plan xOz (where $y = 0$) and representation of the lidar measurements for the wind reconstruction ahead of a plane b) Evolution of the vertical wind component as a function of the position in the simulated wind volume. In black the actual wind component on the flight path, in red the wind component retrieved with an angle of $15°$, in green with an angle of $50°$

a 3 by 3 symmetric matrix $W(k) = (\Phi_{ij}(k))_{i=[x,y,z],j=[x,y,z]}$ is obtained, which is factorized using a Cholesky decomposition to facilitate the generation of correlated random wind. This matrix is then multiplied by a vector of 3 random phases following

a reduced centered normal distribution for each $k$. The three correlated wind components are obtained with an inverse Fourier transformation according to $k$. Note that the temporal evolution of the wind is not taken into account to simplify the study, assuming that the wind does not vary significantly as the plane moves through the grid. Once the winds are simulated, the plane traverses the grid with the lidar in the nose, and the wind is reconstructed in front of it with the same measurement geometry as before.

For the simulation, the wind box was taken equal to $8\,\mathrm{km} \times 800\,\mathrm{m} \times 800\,\mathrm{m}$, sampled every $5\,\mathrm{m}$ in each directions.The turbulence length scale is taken equal to $762\,\mathrm{m}$, value at $10\,\mathrm{km}$ of altitude, and we assume a turbulence with a standard deviation of the wind amplitude of $10\,\mathrm{m\,s^{-1}}$. We considered that the plane is moving at $V_{aircraft} = 250\,\mathrm{m\,s^{-1}}$, that it is centered at $y = 0$ and $z = 0$, along $x$ axis, and that the lidar is located in the nose of the aircraft. In particular, for each measurement, we account for the plane motion during the integration time, that is to say the slight variation of the measured projected wind observed by

each laser pulse. We used a simplified model for the lidar, considering, at each pulse, only one measurement on the laser axis of the projected wind speed at a range $z = d/\cos(\theta)$ over a range gate of $25\,\mathrm{m}$. The model of the lidar measurement noise is assumed to be Gaussian, with the projected wind speed obtain on the range gate for the mean and a standard deviation corresponding to $\sigma_{\mathrm{lidar}}$. For the simulation we take the one obtained for the Merion C. Once the wind components are estimated, the RMSE is estimated at each simulator run, using the mean value of the squared differences between the estimated wind

component and the true wind component over the flight path over 312 values ($\approx 8\,\mathrm{km}/25\,\mathrm{m}$), and taking the square root of the resulting value. Figure 6 displays the results. In Fig. 6.a), an example of wind simulated with the Von Karman model is

depicted. In Fig. 6.b), the green line, that corresponds to the vertical component retrieved using a lidar angle of $50°$, is closer to the black line, that represents the real wind, than the red line, that corresponds to the vertical component reconstructed using a lidar angle of $15°$. 180 runs of simulations have been performed and statistics on all obtained RMSE shows a mean value of $12.7\,\mathrm{m\,s^{-1}}$ with a $3\sigma$ error of $0.3\,\mathrm{m\,s^{-1}}$ for the angle of $15°$ and a mean value of $7.2\,\mathrm{m\,s^{-1}}$ with a $3\sigma$ error of $0.15\,\mathrm{m\,s^{-1}}$ for the angle of $50°$. This illustrates the improvement achieved with the optimized lidar angle of $50°$. In addition, the result is close to the theoretical RMSE given in Fig. 5b). This shows that the motion of the plane during the integration time (i.e. 25 m) has little effect on the error in the wind reconstruction.

## 4 Conclusions

A robust UV lidar architecture, including a QMZ interferometer and a fiber laser, was presented for measuring wind in the feed-forward GLA system. The end-to-end simulator was described and utilized to optimize the lidar architecture. The transmitter/receiver configuration was optimized to ensure complete overlap of the laser and telescope field of view between $100\,\mathrm{m}$ and $300\,\mathrm{m}$. PMTs were chosen due to their high gain, which helps limit the impact of detection noise. An optimized solar filter size was estimated for each laser configuration to mitigate the background signal. Three lasers were selected: a commercial laser, the Merion C, for initial testing, and two fiber laser models studied at ONERA. The error on the projected wind speed estimated at $150\,\mathrm{m}$ on the laser axis are $0.17\,\mathrm{m\,s^{-1}}$ for the Merion C, $0.16\,\mathrm{m\,s^{-1}}$ for the fiber laser, and $0.09\,\mathrm{m\,s^{-1}}$ for the hybrid fiber laser. The simulations focused on GLA application, but the simulator can be applied to other lidar performing wind measurement from space or in High altitude platform. In particular, we used previously this simulator to calculate lidar performances for wind measurement from space using the same architecture as Aeolus, but replacing the laser by a theoretical UV fiber laser and the two spectral analyzer by one QMZ interferometer (Boulant et al., 2023).

The lidar is being assembled and the first validation will be performed soon. All the instrument characteristics (different noise levels, instrumental transmission and so on) will be performed and compared with simulation. In particular, the contrast of the different channel and the phase differences between channels of the QMZ will be measured and simulated to evaluate their effect on the lidar performances to refine the calculation of the performances of the system. Currently we assume a perfect interferometer with all transmission of the optics equal to 1 and an instrumental contrast of 1. This will be reevaluated in future studies with transmission and contrast measured experimentally.

A lidar angle optimization method was presented. This method evaluates the RMSE between the reconstructed vertical wind component and the actual one on the flight path for a linear least squares method. Two contributions are considered, instrumental noise and noise induced by turbulence. The angle that minimizes the RMSE in the presence of Von Karman turbulence is approximately $50°$, resulting in an RMSE that is approximately $50\,\%$ of the RMSE obtained with an angle of $15°$-$30°$. The method has been validated through simulations of turbulence and wind reconstruction. It should be noted that this method can also be applied by considering only instrumental noise. Additionally, this method demonstrates that the intuition of using a small lidar angle to maintain almost homogeneous wind field conditions between measurement points to minimize error on the vertical component is misleading. Indeed, the error between the projections on the lidar axes and those at the

point of reconstruction, induced by turbulence, are amplified by the factor $\frac{1}{\tan(\theta)}$ for small angle. In a future work, we plan
to validate experimentally the improvement of the precision on the reconstructed 3D wind with an existing heterodyne wind
lidar at ONERA. The lidar will point measure the 3D wind along a central axis using four axis evenly distributed around
this central axis. The reconstructed wind will be compared with the "true" wind measured with an independent local detector
(anemometer). This comparison will be performed for several angles between the four beams and the central axis to validate
this calculated improvement.

The effect of refraction due to turbulence has not been taken into account in the 3D wind simulator. In the UV, the refraction
is strong and the beam can be significantly deflected as it propagates through the atmosphere. This can lead to an increase in
the size of the probe volume, depending on the direction of the refraction. This will be addressed in future studies. In addition,
the evolution of the RMSE with multiple runs of the simulator will be performed in future work to evaluate the sensitivity of
RMSE with different turbulence.

The lidar angle have been optimized, but the simulator shows that for high strength turbulence, the error on the reconstructed
wind is still high ($7.4\,\mathrm{m\,s^{-1}}$ with an angle of $50°$) and not allow to reach the GLA requirement of $1\,\mathrm{m\,s^{-1}}$. This is the limit
of the estimator with this measurement geometry. To be noted that further reconstruction method are studied at ONERA to
decrease the error on the reconstructed wind speed.

## Appendix A:  Lidar measurement standard deviation with Speckle noise

For the calculation of the speckle noise contribution on standard deviation, we use the method describe by Bruneau and Pelon
(2003)

### A1    estimation of the projected wind speed from the four currents

At the output of the four detectors, if we assume that there is the same transmision at the four outputs of the QMZ and the
interferometer introduces no loss of contrast, the signals in current are :

$$S_1 = \frac{S_0}{4}(1 + M_{\mathrm{tot}}\cos(\varphi)) + S_{\mathrm{bkg}} + S_{\mathrm{dark}} \tag{A1}$$

$$S_2 = \frac{S_0}{4}(1 - M_{\mathrm{tot}}\sin(\varphi)) + S_{\mathrm{bkg}} + S_{\mathrm{dark}} \tag{A2}$$

$$S_3 = \frac{S_0}{4}(1 - M_{\mathrm{tot}}\cos(\varphi)) + S_{\mathrm{bkg}} + S_{\mathrm{dark}} \tag{A3}$$

$$S_4 = \frac{S_0}{4}(1 + M_{\mathrm{tot}}\sin(\varphi)) + S_{\mathrm{bkg}} + S_{\mathrm{dark}} \tag{A4}$$

where $S_0$ the total current, $M_{\mathrm{tot}}$ the contrast of the interferences due to the spectral shape of the input light, $S_{\mathrm{bkg}}$ the signal due
to the background signal, $S_{\mathrm{dark}}$ the noise from the detector and $\varphi$ a simpler notation of $\delta\varphi_{\mathrm{OPD}}$. The spectrum of the incident
light is from two scattering process, Rayleigh scattering and Mie scattering, so $M_{\mathrm{tot}} = \frac{1}{R_\beta}M_m + \frac{R_\beta - 1}{R_\beta}M_p$ with $R_\beta$ is the
backscatter ratio, $M_m$ the interference contrast due to the Rayleigh spectrum and $M_p$ the interference contrast due to the Mie
spectrum.

Considering that the background noise and the detection noise can be estimated and subtracted to the signals, $\varphi$ can be estimated by :

$$Q_1 = \frac{S_1' - S_3'}{S_1' + S_3'} = M_{\text{tot}} \cos(\varphi) \tag{A5}$$

$$Q_2 = \frac{S_4' - S_2'}{S_4' + S_2'} = M_{\text{tot}} \sin(\varphi) \tag{A6}$$

$$\varphi = \arctan(\frac{Q_2}{Q_1}) \tag{A7}$$

where $S_i'$ indicates that background and the detection signals have been subtracted to the signals $S_i$. Doing the same with the reference signal (i.e the laser light), we obtain $\varphi_0$ and the projected wind speed is estimated with:

$$v_p = \frac{\lambda_0 c}{4\pi \Delta L}(\varphi - \varphi_0) \tag{A8}$$

**A2   standard deviation on the wind speed due to the Speckle noise**

Let's note $Q = \frac{Q_1}{Q_2}$ and $S_\varphi = \frac{4\pi \Delta L}{\lambda_0 c}$. The standard deviation of $v_p$ is (neglecting the standard deviation on $\varphi_0$ because the reference signal is sufficiently high to have a good SNR):

$$\sigma_{v_p} = \frac{1}{S_\varphi}\sigma_\varphi \tag{A9}$$

where $\sigma_\varphi$ is the standard deviation on $\varphi$ :

$$\varphi = \frac{d\varphi}{dQ}\sigma_Q = \cos(\varphi)^2\sigma_Q \tag{A10}$$

with $\sigma_Q$ the standard deviation on Q. The variance on Q is (as $Q_1$ end $Q_2$ are uncorrelated):

$$\text{var}(Q) = Q^2(\frac{\text{var}(Q_1)}{Q_1^2} + \frac{\text{var}(Q_2)}{Q_2^2})) \tag{A11}$$

Taking the expression (A5) we have :

$$\text{var}(Q_1) = Q_1^2(\frac{\text{var}(S_1' - S_3')}{(S_1' - S_3')^2} + \frac{\text{var}(S_1' + S_3')}{(S_1' + S_3')^2} - 2\frac{\text{cov}(S_1' - S_3', S_1' + S_3')}{(S_1' - S_3')(S_1' + S_3')}) \tag{A12}$$

Assuming that $S_1'$ and $S_3'$ are uncorrelated, we have:

$$\text{var}(S_1' - S_3') = \text{var}(S_1' + S_3') = \text{var}(S_1') + \text{var}(S_3') \tag{A13}$$

$$\text{cov}(S_1' - S_3', S_1' + S_3') = \text{var}(S_1') - \text{var}(S_3') \tag{A14}$$

The variance of $Q_1$ is:

$$\text{var}(Q_1) = (1 + Q_1^2)\frac{\text{var}(S_1') + \text{var}(S_3')}{(S_1' + S_3')^2} - 2Q_1\frac{\text{var}(S_1') - \text{var}(S_3')}{(S_1' + S_3')^2} \tag{A15}$$

In the case where the Speckle noise dominates we have $\text{var}(S_1') = \frac{S_{1m}'^2}{N_m} + \frac{S_{1p}'^2}{N_p}$ with $S_{1m}' = \frac{S_{0m}}{4}(1 + M_m \cos(\varphi))$ and $S_{1p}' = \frac{S_{0p}}{4}(1 + M_p \cos(\varphi))$ where $S_{0m}$ and $S_{0p}$ are the intensities at the input of the interferometer coming from molecules and

450 particle respectively. Same thing for $\text{var}(S_3') = \frac{S_{3m}'^2}{N_m} + \frac{S_{3p}'^2}{N_p}$ with $S_{3m}' = \frac{S_{0m}}{4}(1 - M_m\cos(\varphi))$ and $S_{3p}' = \frac{S_{0p}}{4}(1 - M_p\cos(\varphi))$.

$N_m$ and $N_p$ are the number of speckle patterns obtained for a given range gate, linked to the size of the laser beam over the scattering volume, and the number of temporal speckles, due to the coherence of the scattered light. The number of spatial pattern is $(\frac{\pi\theta_{\text{div}}r_{\text{pup}}}{2\lambda})^2$ where $\theta_{\text{div}}$ is the half divergence of the laser beam and $r_{pup}$ the radius of the telescope pupil (Goodman, 1975). The number of temporal speckle pattern is $\frac{2\delta z}{c\tau_{\text{coh}}}$ with $\delta z$ the range gate and $\tau_{coh}$ the coherence length of the signal

(Cezard, 2008). The coherence length is inversely proportional to spectrum width. Therefore, as the spectrum of the Rayleigh signal is wider than that of the Mie signal due to a larger Boltzmann distribution, the number of time patterns will be higher for Rayleigh than for Mie.

We obtain for the variance of $Q_1$:

$$\text{var}(Q_1) = (1+Q_1^2)\frac{\frac{(S_{0m}/4)^2}{N_m}(2 + 2M_m^2\cos(\varphi)^2) + \frac{(S_{0p}/4)^2}{N_p}(2 + 2M_p^2\cos(\varphi)^2)}{(S_0/2)^2}$$

$$- 2Q_1\frac{\frac{(S_{0m}/4)^2}{N_m}4M_m\cos(\varphi) + \frac{(S_{0p}/4)^2}{N_p}4M_p\cos(\varphi)}{(S_0/2)^2} \tag{A16}$$

We can see that two contributions appear in the expression: the one from the Rayleigh signal and the one of the Mie signal. In the following, we only made the calculation for the Rayleigh signal and consider that the calculation are the same for the Mie signal. If we note $Q_{1m} = M_m\cos(\varphi)$ and $Q_{1p} = M_p\cos(\varphi)$, the variance of $Q_1$ is:

$$\text{var}(Q_1) = \frac{\frac{(S_{0m}/4)^2}{N_m}(2 + 2Q_{1m}^2) + \frac{(S_{0m}/4)^2}{N_m}(2Q_1^2 + 2Q_1^2 Q_{1m}^2) - 8Q_1 Q_{1m}\frac{(S_{0m}/4)^2}{N_m} + (\text{Mie part})}{(S_0/2)^2}$$

$$= (\frac{S_{0m}}{S_0})^2\frac{1}{2N_m}[(1 - Q_1 Q_{1m})^2 + (Q_1 - Q_{1m})^2] + (\frac{S_{0p}}{S_0})^2\frac{1}{2N_p}[(1 - Q_1 Q_{1p})^2 + (Q_1 - Q_{1p})^2] \tag{A17}$$

For the variance of $Q_2$, the calculation is the same:

$$\text{var}(Q_2) = (\frac{S_{0m}}{S_0})^2\frac{1}{2N_m}[(1 - Q_2 Q_{2m})^2 + (Q_2 - Q_{2m})^2] + (\frac{S_{0p}}{S_0})^2\frac{1}{2N_p}[(1 - Q_2 Q_{2p})^2 + (Q_2 - Q_{2p})^2] \tag{A18}$$

with $Q_{2m} = M_m\sin(\varphi)$ and $Q_{2p} = M_p\sin(\varphi)$. Considering that the calculation are the same for the Rayleigh part and the Mie part of the formula, we have:

$$\frac{\text{var}(Q_1)}{Q_1^2} + \frac{\text{var}(Q_2)}{Q_2^2} = (\frac{S_{0m}}{S_0})^2\frac{1}{2N_m}[\frac{(1 - Q_1 Q_{1m})^2 + (Q_1 - Q_{1m})^2}{Q_1^2} + \frac{(1 - Q_2 Q_{2m})^2 + (Q_2 - Q_{2m})^2}{Q_2^2}] + (\text{Mie part})$$

$$= (\frac{S_{0m}}{S_0})^2\frac{1}{2N_m}\frac{M_{\text{tot}}^2[1 + (M_{\text{tot}}^2 M_m^2 - 4M_{\text{tot}}M_m + 2(M_{\text{tot}} - M_m)^2)\frac{\sin(2\varphi)^2}{4}]}{M_{\text{tot}}^4\cos(\varphi)^2\sin(\varphi)^2} + (\text{Mie part}) \tag{A19}$$

Finally, the variance of Q is :

$$\text{var}(Q) = (\frac{S_{0m}}{S_0})^2\frac{1}{2N_m M_{\text{tot}}^2\cos(\varphi)^2}[1 + (M_{\text{tot}}^2 M_m^2 - 4M_{\text{tot}}M_m + 2(M_{\text{tot}} - M_m)^2)\frac{\sin(2\varphi)^2}{4}]$$

$$+ (\frac{S_{0p}}{S_0})^2\frac{1}{2N_p M_{\text{tot}}^2\cos(\varphi)^2}[1 + (M_{\text{tot}}^2 M_p^2 - 4M_{\text{tot}}M_p + 2(M_{\text{tot}} - M_p)^2)\frac{\sin(2\varphi)^2}{4}] \tag{A20}$$

If we assume that atmospheric transmission are almost equal to 1, $\frac{S_{0m}}{S_0} \approx \frac{1}{R_\beta}$ and $\frac{S_{0p}}{S_0} \approx \frac{R_\beta - 1}{R_\beta}$. The variance on the projected wind due to the Speckle noise is:

$$
\begin{aligned}
\sigma_{v_p}^2 = &\left(\frac{1}{R_\beta}\right)^2 \frac{1}{2 N_m M_{\text{tot}}^2 S_\varphi^2}\left[1 + (M_{\text{tot}}^2 M_m^2 - 4 M_{\text{tot}} M_m + 2(M_{\text{tot}} - M_m)^2)\frac{\sin(2\varphi)^2}{4}\right] \\
&+ \left(\frac{R_\beta - 1}{R_\beta}\right)^2 \frac{1}{2 N_p M_{\text{tot}}^2 S_\varphi^2}\left[1 + (M_{\text{tot}}^2 M_p^2 - 4 M_{\text{tot}} M_p + 2(M_{\text{tot}} - M_p)^2)\frac{\sin(2\varphi)^2}{4}\right]
\end{aligned}
\tag{A21}
$$

### Appendix B: Expression of the MSE due to the turbulence

For the different wind, we will note $V = (V_x, V_y, V_z)$ the wind on the flight path at range $d$, $V_1 = (V_{x1}, V_{y1}, V_{z1})$ the wind located at the measurement point of axis 1 and $V_3 = (V_{x3}, V_{y3}, V_{z3})$ the wind located at the measurement point of axis 3. Then $\delta V_{p1} = (V_{x1} - V_x)\cos(\theta) - (V_{z1} - V_z)\sin(\theta)$ and $\delta V_{p3} = (V_{x3} - V_x)\cos(\theta) + (V_{z3} - V_z)\sin(\theta)$. The variance of $\delta V_{p1}$ is:

$$
\langle \delta V_{p1}^2 \rangle = \langle (V_{x1} - V_x)^2 \rangle \cos(\theta)^2 + \langle (V_{z1} - V_z)^2 \rangle \sin(\theta)^2 - 2\sin(\theta)\cos(\theta)\langle (V_{x1} - V_x)(V_{z1} - V_z)\rangle
\tag{B1}
$$

   $\langle (V_{x1} - V_x)^2 \rangle$ and $\langle (V_{z1} - V_z)^2 \rangle$ correspond to the lateral structure function and longitudinal structure function respectively.
The paper of Wilson (1998) gives the formula for the correlation between two wind component : $\langle V_i V_j \rangle(r) = (B_{LL}(r) - B_{NN}(r))\frac{r_i r_j}{r^2} + B_{NN}(r)\delta_{ij}$ with $\delta_{ij}$ the Kronecker symbol, $r_i$ and $r_j$ the components $i$ and $j$ of the displacement vector $r$. As $r_x = 0$, the correlation between two wind components of different directions with one of them equal to $x$ direction is equal to 0, that led to:

$$
\langle \delta V_{p1}^2 \rangle = D_{NN}(r)\cos(\theta)^2 + D_{LL}(r)\sin(\theta)^2
\tag{B2}
$$

with $D_{LL}(r) = 2(B_{LL}(0) - B_{LL}(r))$. By symmetry, $\langle \delta V_{p1}^2 \rangle = \langle \delta V_{p3}^2 \rangle$. For the correlation between the two differences, we have:

$$
\begin{aligned}
\langle \delta V_{p1} \delta V_{p3} \rangle &= \langle ((V_{x1} - V_x)\cos(\theta) - (V_{z1} - V_z)\sin(\theta))((V_{x3} - V_x)\cos(\theta) + (V_{z3} - V_z)\sin(\theta))\rangle \\
&= \cos(\theta)^2 \langle (V_{x1} - V_x)(V_{x3} - V_x)\rangle - \sin(\theta)^2 \langle (V_{z1} - V_z)(V_{z3} - V_z)\rangle \\
&= \cos(\theta)^2 (B_{NN}(2r) + D_{NN}(r) - B_{NN}(0)) - \sin(\theta)^2 (B_{LL}(2r) + D_{LL}(r) - B_{LL}(0))
\end{aligned}
\tag{B3}
$$

Here again, passage from B3 to B4 is due to the correlation between two wind components of different directions with one of them equal to $x$ direction is equal to 0. Using equations B1 and B5 in the expression of $MSE_T|_{\text{turbulence}}$, we obtain:

$$MSE_T|_{\text{turbulence}} = \frac{2}{(2\sin{(\theta)})^2} \left( \langle \delta V_{p1}^2 \rangle + \langle \delta V_{p3}^2 \rangle - 2 \langle \delta V_{p1} \delta V_{p3} \rangle \right)$$

$$= \frac{D_{NN}(2r)}{(2\tan{(\theta)})^2} + \frac{3B_{LL}(0) + B_{LL}(2r)}{2} - 2B_{LL}(r) \tag{B4}$$

$$\tag{B5}$$

**Table A1.** First table that summarizes the parameters of the lidar and the simulation

| Reception telescope | |
|---|---|
| diameter | 152.4 mm (6 in) |
| Focale length | 609.6 mm (24 in) |
| Aperture | f/4 |
| Secondary mirror diameter | 38 mm |
| *Focusing distance* | 155 m |
| **Fiber** | |
| Core diameter | 400 μm |
| Numerical aperture | 0.22 |
| **Laser** | |
| Beam size after emission telescope | 30 mm |
| $M^2$ | <8 |
| *Beam waist position* | 100 m |
| Spectral width $(1/e^2)$ | 400 MHz |
| Pulse duration | 10 ns |
| *Merion C* | |
| Pulse energy | 22.5 mJ |
| PRF | 400 Hz |
| *Fiber Laser* | |
| Pulse energy | 250 μJ |
| PRF | 40 kHz |
| *Hybrid fiber Laser* | |
| Pulse energy | 750 μJ |
| PRF | 40 kHz |
| **Solar filter** | |
| *Bandwidth* | 1 nm |
| **Background** | |
| Background radiance | $0.3\,\mathrm{W\,m^{-2}\,sr^{-1}\,nm^{-1}}$ |

Parameters in italics are those deduced from simulations

**Table B1.** Second table that summarizes the parameters of the lidar and the simulation

| **Detector SiPIN S5971 Hamamatsu** | |
| --- | --- |
| Quantum efficiency | 0.5 |
| Gain | 1 |
| Noise factor | 1 |
| Dark current | 0.07 nA |
| **Detector SiAPD S9075 Hamamatsu** | |
| Quantum efficiency | 0.5 |
| Gain | 5 |
| Noise factor | 1.57 |
| Dark current | 0.5 nA |
| *Detector PMT R10721-210 Hamamatsu* | |
| Quantum efficiency | 0.43 |
| Gain | $2 \times 10^6$ |
| Noise factor | 1.3 |
| Dark current | 10 nA |
| **Atmosphere** | |
| Particle backscattering coefficient (<10 km) | $8 \times 10^{-6}\,\mathrm{m}^{-1}\,\mathrm{sr}^{-1}$ |
| **Measurement parameters** | |
| Range gate | 25 m |
| Measurement time | 0.1 ms |

Parameters in italics are those deduced from simulations

*Author contributions.* Authors equally contributed to the writing of this paper

*Competing interests.* The authors declare that their is no competing interests

*Acknowledgements.* The authors wish to acknowledge Nicolas Cézard and Anasthase Liméry from ONERA for their help to the development of the simulation code of the emission/reception telescope.

    This study is funded by the DGAC

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
