# Peer review of "Optimization of a direct detection UV wind lidar architecture for 3D wind reconstruction at high altitude"

_Atmospheric Measurement Techniques, 2024_

## Referee Comment (RC3)

***Comment on*** **"Optimization of a direct detection UV wind lidar architecture for 3D wind reconstruction at high altitude," by Thibault Boulant, Tomline Michel, and Matthieu Valla**

This manuscript describes an investigation that is relevant to Atmospheric Measurement Techniques in the field of remote sensing technique regarding an airborne UV Doppler Wind Lidar (DWL). The authors describe a study for the architecture of a UV molecular lidar designed to make the lateral and vertical wind measurement in front of an aircraft for Gust Load Alleviation (GLA) applications and optimization of performance on wind measurement. The manuscript is interesting approach to optimize performance on the lateral and vertical wind measurements of the UV airborne DWL. The authors led to the results of both wind measurement errors for the three laser systems and the optimum angle between the direction of laser beam and the airplane axis. I believe that the manuscript will be of interest to the readers of Atmospheric Measurement Techniques. However, regarding the content of the manuscript, there are issues that should be addressed prior to publication:

General comment

The authors assumed to be the measurement time of 0.1 sec, which corresponds to a range resolution of 25m for an aircraft speed of 250 m/sec. The target measurement range is 100 to 150 m. First concerning is the spatiotemporal and accuracy requirements of the wind measurement for the GLA. What is the response speed meeting the requirements? Please add explanation and references for the requirements.

Second concern is the signal to noise ratio (SNR) at the focusing region at the lidar angles $\theta$ of 10 and 50 degrees. The backscatter coefficient at an altitude of 10 km is not shown in the manuscript. The size for the laser beam and receiver filed at the focusing plane is not shown. What is the SNR and the detectability required for the GLA?

Regarding to the second concern, third concern is repetitiveness of the wind filed. The $\theta$ of 10 and 50 degrees have different volumes. When the laser beam and the receiver filed are focused, the reviewer thinks that the wind filed at the focusing volume will be no longer the repetitiveness around the plane. The refractive turbulence is strong in the UV.

Specific Comments
1. Line 17: Introduction. Please add references regarding to aviation accident and safety.
2. Lines 26-27: "100m- 200m ahead of the aircraft" is the spatial requirement. Please add explanation and reference related to spatial requirement.
3. Lines 31 and 88: "abbreviation for Gust Load Alleviation" is shown again. Please remove "Gust Load Alleviation" if you use the abbreviation GLA.
4. Line 55: QMZ should be "Quadri Mach-Zehnder (QMZ)".
5. Lines 62-63: "a factor $\sqrt{2}$ increase in statistical error" is not clear. Please add explanation regarding on the factor and the reference.
6. Line 110: How much is the numerical aperture (NA) of multimode fiber assumed in the manuscript?
7. Line 156: What is "moy"?

8. Line 162: "Maximum Likelihood Estimator (MLE)".   Please add reference.

9. Line 165: "Cramer Rao's lower bound".   Please add reference.

10. Lines 168-169: The laser beam shown in Figure 2(a) should be convergent.

11. Lines 170-171: Please add explanation regarding on relation between F-number and the NA.

12. Line 172: I don't understand $M^2<8$.   Please add explanation regarding on physical meanings of $M^2$ and $M^2<8$.

13. Line 190: What is "$\gamma(r)2$"?   Please add explanation.   Is "$\gamma(r)^2$" correct?

14. Lines 194, 236, and 240: "0," should be "0.".

15. Lines 199-200: The backscatter coefficient at an altitude of 10 km is not shown in the manuscript.   Please add the backscatter coefficient.

16. Line 200: "$m^{-1}.sr^{-1}$" should be "/m/sr".

17. Line 203: Laser has a spectral width of <500 MHz.   How long is the pulse width assumed?   Is it possible to develop the laser system?

18. Line 204: Please add references regarding the spectral broadening of 3 GHz.

19. Line 207: Figure 3 shows result of the relation between laser average power and pulse repetition frequency (Hz).   Do you need to show the results at the laser average power of >10W operating at PRF of < 100 Hz. Is it feasible to develop the laser system in the term of the laser power density and laser-induced optical damage?

20. Line 230: "repetition rate of 400 Hz and delivers 22.5 mJ of energy per pulse".   In Figure 3, Merion C is "9W 40 kHz".   Which is it correct?

21. Lines 225-240: Do three lasers have enough tolerance for the optical damage?   Does each laser have enough tolerance for the optical damage?

22. Figure 4: Please embed "high altitude" and "low altitude" int the Figures 4(a) and 4(b), respectively.

23. Line 270: "root mean square error". "root mean square error (RMS)" should be better.

24. Line 297: "root mean square error" should be removed.

25. Line 325: "Figure 5) displays the results.".   ")" should be deleted.   "results", how did you simulate?   It is not clear.   Please add explanation.   Did you investigate the statistic difference between real wind component and retrieved wind component at two angles of 15 and 50 degree?   Please add results and explanation.

Refences

26. Line 445.   What is the title?   Which journal is the manuscript reviewed?

27. Line 459.   2022 -> 2021.

Miscellaneous

28. The summary of specification parameters used in the simulations will be helpful for the readers. Please add the summary of the specification parameters.

---

## Author Comment (AC1)

Referee comments 1:

This paper focuses on two issues associated with application of a direct-detection Doppler lidar for measuring winds ahead of an aircraft to feed forward for gust load alleviation. First, it describes development of a design to maximize wind measurement precision, focusing on telescope, background filter, and detector specifications. It then utilizes an atmospheric model of molecular and aerosol backscatter to predict performance for three different laser transmitters. The second issue addressed is optimization of the angle of deviation for a lidar that utilizes a four-direction concept, assuming turbulence from a commonly employed turbulence model.

I find the paper quite well done and certainly worthy of publication. Although the two main issues could have been addressed separately, they fit together acceptably into a single article. The figures are appropriate and illustrate the main points and conclusions.

In reading the paper, I would have liked to have seen a bit more discussion of the QMZ interferometer and explanation of how the SNR lead to errors in wind speed. This is more of a personal preference – the paper is very well referenced and the Appendix provides the necessary details for estimating wind error from photon count. Perhaps a figure that illustrates how the output from the detectors changes as function of wind speed (or phase) would be tutorial and useful in illustrating the concept.

Although speckle noise is an important component in velocity measurement uncertainty estimate, there is very little discussion of the basis for speckle noise and which system and laser transmitter parameters affect it. For example, line 223 on page 8 says that "the speckle noise decreases because the backscattering is predominantly molecular, which is less coherent than particulate backscattering. I may have missed it, but I didn't see in the text or the appendix that discusses the speckle relative to the signal coherence and how this is incorporated into the simulation.

Figure 3 is quite informative and sums up the discussion on wind speed standard deviation nicely.

The angle optimization part of the study produces a nice and useful result. While reviewing the paper, I thought that thus problem had to have been addressed earlier in slider studies of wind energy, but I perused the literature a bit and didn't find it. Consequently, this result should be of significant interest to the community.

As with all simulation studies, this work begs for follow-up research to demonstrate the concept and validate the simulation. The authors should add some text at the end on anticipated future work and tell the reader how they intend to use the results of the study.

General answer; We would like to thank the referee 1 for showing a keen interest in our article and in particular for finding the paper "quite well done and certainly worthy of publication". All his comments have been addressed in the following and the paper have been modified in this regard. We have also added small additional modifications throughout the paper to improve its quality. Please consider this revised manuscript for publication in AMT.

**Referee 1: In reading the paper, I would have liked to have seen a bit more discussion of the QMZ interferometer and explanation of how the SNR lead to errors in wind speed. This is more of a personal preference – the paper is very well referenced and the Appendix provides the necessary details for estimating wind error from photon count. Perhaps a figure that illustrates how the output from the detectors changes as function of wind speed (or phase) would be tutorial and useful in illustrating the concept.**

Answer: We agree with the referee and we have done the following changes: (end 2nd paragraph section 2.1) "…which is then sampled and digitized into a computer. Figure 3 shows the evolution of the simulated signal at the output of the detectors, for the reference signal and the Rayleigh signal. Signal processing makes it possible to recover the phase of the interference…"

(see Figure 3 in the revised version)

The variation of SNR can leads to errors in wind speed estimation because wind speed estimation implies the ratio of the difference between signals from detectors to the sum of the signals. This means that, in the variance of the wind speed, the ratio of the variance of signals to the sum of signals will appear (see eq A15), which is homogeneous to the inverse of a SNR. This mean that when the SNR increases, the standard deviation decreases.

**Referee 1: Although speckle noise is an important component in velocity measurement uncertainty estimate, there is very little discussion of the basis for speckle noise and which system and laser transmitter parameters affect it. For example, line 223 on page 8 says that "the speckle noise decreases because the backscattering is predominantly molecular, which is less coherent than particulate backscattering. I may have missed it, but I didn't see in the text or the appendix that discusses the speckle relative to the signal coherence and how this is incorporated into the simulation.**

Answer: We agreed with the referee and added:

- After equation (A15) in appendix A, line 432: "N_m and N_p are the number of speckle patterns obtained for a given range gate, linked to the size of the laser beam over the scattering volume, and the number of temporal speckles, due to the coherence of the scattered light. The number of spatial pattern is $\left(\frac{\pi\theta_{div}r_{pup}}{2\lambda}\right)^2$ where $\theta_{div}$ is the half divergence of the laser beam and $r_{pup}$ the radius of the telescope pupil. The number of temporal Speckle pattern is $\frac{2\delta z}{c\tau_{coh}}$ with $\delta z$ the range gate and $\tau_{coh}$ the coherence length of the signal. The coherence length is inversely proportional to spectrum width. Therefore, as the spectrum of the Rayleigh signal is wider than that of the Mie signal due to a larger Boltzmann distribution, the number of time patterns will be higher for Rayleigh than for Mie."

- How it is incorporated into the simulation (section 2.3.4, line 222): ".... We considered that the laser has a full width at $1/e^2$ of 400 MHz, significantly less than the spectral broadening induced by the thermal movement of the molecules (6.3 GHz for a full width at $1/e^2$). For the Mie scattering, the coherence time is limited by the laser pulse duration, i.e. 10 ns. For the Rayleigh scattering, it is limited by spectral broadening due to thermal motion of the molecule, i.e. 0.63 ns for a broadening of 6.3 GHz at $1/e^2$. The simulations were conducted both on the ground and at 10 km altitude, approximately corresponding to the aircraft's cruising altitude, as the GLA must operate throughout the flight.

**Referee 1: Figure 3 is quite informative and sums up the discussion on wind speed standard deviation nicely.**

We appreciate the comments

**Referee 1: The angle optimization part of the study produces a nice and useful result. While reviewing the paper, I thought that thus problem had to have been addressed earlier in slider studies of wind energy, but I perused the literature a bit and didn't find it. Consequently, this result should be of significant interest to the community**.

We did not find either studies published previously where lidar addressing angle was optimized for 3D wind reconstruction. We appreciate very much the comment.

**Referee 1: As with all simulation studies, this work begs for follow-up research to demonstrate the concept and validate the simulation. The authors should add some text at the end on anticipated future work and tell the reader how they intend to use the results of the study.**

Answer: We propose to modify the conclusion as follow (line 379) : "…Additionally, this method demonstrates that the intuition of using a small lidar angle to maintain almost homogeneous wind field conditions between measurement points to minimize error on the vertical component is misleading. Indeed, the error between the projections on the lidar axes and those at the point of reconstruction, induced by turbulence, are amplified by the factor $\frac{1}{2\tan(\theta)}$ for small angle. In a future work, we plan to validate experimentally the improvement of the accuracy on the reconstructed 3D wind when increasing the addressing angle with an existing heterodyne wind lidar at ONERA. The lidar will be used to reconstruct the 3D wind along a central axis using four axis evenly distributed around this central axis. The reconstructed wind will be compared with the "true" wind measured with an independent local detector (anemometer); This comparison will be performed for several angles between the four beams and the central axis to validate this calculated improvement."

---

## Author Comment (AC2)

Answer: We would like to thank referee 2 for reading the article thoroughly, and for the detailed corrections that he suggests to improve the paper's impact on the scientific community. We appreciate that referee 2 says "The work is certainly of interest to readers working in the field of active remote sensing, and particularly to the wind lidar community", that he finds the article "well-structured" and that the "figures support the intriguing findings derived from the simulations". All his comments have been addressed in the following and the paper have been modified in this regard. We have also added small additional modifications throughout the paper to improve its quality. Please consider this revised manuscript for publication in AMT.

Please find bellow more detailed response to the referee 2:

**Anonymous Referee : However, given the results reported in the paper, I am wondering a) to what extent the developed end-to-end simulator is applicable to other wind lidar instruments based on a QMZ (not necessarily for GLA), b) what the major limitations of the simulator are, c) what refinements of the simulator are planned in the future, and d) if (or when) experimental validation of the simulation results with the described instrument is foreseen. In my opinion, these four relevant questions are currently unaddressed and should be discussed in the conclusions section of the manuscript.**

**a) to what extent the developed end-to-end simulator is applicable to other wind lidar instruments based on a QMZ (not necessarily for GLA)**

Answer:

a) Concerning the simulation of the atmosphere, the model can be extended to any simulation below 20 km of altitude as the molecular distribution is describes by the US standard atmosphere model. Concerning the Mie scattering, the difficulty is to model correctly the backscattering coefficient as it change with time/location/ altitude and so on. In our model we assumed beta = 8e-6 $m^{-1}.str^{-1}$ under 1 km of altitude and 0 above. This value is the median for the particle backscattering coeficient at 355 nm according to (Herbst, 2016). This could be refined.

The code for the simulation of the telescope (calculation of the overlap function) can also be extended to other telescope configuration. For example, the study (Boulant et al., 2023) present the work we have done to modify the architecture for space based wind measurement.

We also use the simulator to optimize an Aeolus-like lidar that use UV fiber laser and a QMZ interferometer as spectral analyzer. In conclusion, we propose to add line 365:

"The simulations focused on GLA application, but the simulator can be applied to other QMZ lidar configuration performing wind measurement from space or in High altitude platform. In particular, we used previously this simulator to calculate lidar performances for wind measurement from space using the same architecture as Aeolus, but replacing the laser by a theoretical UV fiber laser and the two spectral analyzers of Aeolus by one QMZ interferometer (Boulant et al., 2023)."

**b) what the major limitations of the simulator are, c) what refinements of the simulator are planned in the future,**

The major limitations of the simulator that will be refined latter are the following:

- We assume a transmission of the optics of 50% from the telescope to the QMZ entrance. This will need to be refined with experimental value.
- We assume a perfect interferometer with a contrast of 1. The imperfection of the optics and of the transmission/reflection may reduce the contrast. It may also lead to different transmission and contrast on each arm. This will be taken into account in future simulations with experimental values of transmissions and contrast.

**d) if (or when) experimental validation of the simulation results with the described instrument is foreseen.**

The instrument is currently under development and should be tested soon.

We have added this paragraph in conclusion, after the first paragraph :

The lidar is being assembled and the first validation will be performed soon. All the instrument characteristics (different noise levels, instrumental transmission and so on) will be performed and compared with simulation. In particular, the contrast of the different channel and the phase differences between channels of the QMZ will be measured and simulated to evaluate their effect on the lidar performances to refine the calculation of the performances of the system.

**Specific comments:**

1. **Given the multitude of system parameters discussed in the text, I strongly suggest to add a table that summarizes the specifications of the lidar instrument including both the given (or fixed) parameters, such as telescope diameter, primary mirror focal length, detector gain, as well as the derived optimized parameters such as focusing distances, solar filter bandwidth, laser pulse energy, pulse repetition frequency, etc.**

Answer: We agree with the referee and we have added the following table in appendix with a reference line 266 : "The main parameters that have been used in the simulation are sum up in table 1 and 2. The parameters in red correspond to the one that were optimized with simulations."

| Reception Telescope | |
| --- | --- |
| Diameter | 152.4 mm (6 in) |
| Focal length | 609.6 mm (24 in) |
| Aperture | f/4 |
| Secondary mirror diameter | 38 mm |
| Focus distance | 155 M |
| | |
| **Fiber** | |
| Core diameter | 400 µm |
| Numerical aperture | 0.22 |
| | |
| **Laser** | |

| | |
|---|---|
| Beam size at emission | 36 mm |
| $M^2$ | < 8 |
| Beam waist position | 100 m |
| Spectral width $(1/e^2)$ | 400 MHz |
| Pulse duration | 10 ns |

**Merion C**

| | |
|---|---|
| Pulse energy | 22.5 mJ |
| Pulse repetition frequency | 400 Hz |

**Fiber Laser**

| | |
|---|---|
| Pulse energy | 250 µJ |
| Pulse repetition frequency | 40 kHz |

**Hybrid fiber laser**

| | |
|---|---|
| Pulse energy | 750 µJ |
| Pulse repetition frequency | 4 0 kHz |

**Solar filter**

| | |
|---|---|
| Bandwidth | 1 nm |

**Backgorund**

| | |
|---|---|
| Solar Background | 0.3 W/m$^2$/str/nm |

**Detector SiPIN S5971 Hamamatsu**

| | |
|---|---|
| Quantum efficiency | 0.5 |
| Gain | 1 |
| Excess noise factor | 1 |
| Dark current | 0.07 nA |

**Detector SiAPD S9075 Hamamatsu**

| | |
|---|---|
| Quantum efficiency | 0.5 |
| Gain | 5 |
| Excess noise factor | 1.57 |
| Dark current | 0.5 nA |

**Detector PMT R10721-210 Hamamatsu**

| | |
|---|---|
| Quantum efficiency | 0.43 |
| Gain | 2e6 |
| Excess noise factor | 1.3 |
| Dark current | 10 nA |

**Atmosphere**

| | |
|---|---|
| Particle backscattering coefficient (< 1km) | 8e-6 m$^{-1}$.str$^{-1}$ |

**Measurement parameter**

| | |
|---|---|
| Range gate | 25 m |
| Measurement time | 0.1 ms |

2. **Line 71: The impact on micro-vibrations on the frequency of the Aeolus laser is discussed in a more recent publication (Lux et al., AMT, 14, 6305–6333, 2021). Please add this reference to the one already provided (Mondin et al., 2017).**

Answer: We agree with the referee that this publication is particularly interesting for our study as vibrations are a major concern. We have added this reference.

3. **Line 198: Why did the authors assume a solar filter bandwidth of 1 nm, although the spectral width of the Rayleigh signal considering Doppler shifts of +/- 100 m/s accounts for only 1 pm (line 186)? What is the actual limitation for the lower bound of the spectral bandwidth (transmission, price)?**

Answer: Yes the thinner the filter, the more expensive and the lower the transmission but the higher it suppress the background signal. Therefore, the choice of the solar filter is a tradeoff between background suppression transmission and price. In the simulation, we fixed the filter bandwidth to 1 nm because such filter is close to the limit of the technology. However, for each configuration, the filter thickness needs to be refined to determine the filter thickness that can be used for each laser parameter.

Section 2.3.3, last paragraph "… For a filter bandwidth of 1 nm, this results in a minimum laser energy per pulse E_{pmin} =  298 µJ. This filter bandwidth is used for the rest of the simulation, as it is close to the limit of the technology in term of filter thickness."

Section 2.3.4, 1$^{st}$ paragraph "…simulations were performed by adjusting the average laser power and pulse repetition frequency to assess the error in the retrieved wind velocity.  We neglected electrical noise by considering PMT detectors. At low altitudes …"

4. **Lines 233ff.: The maximum solar filter bandwidths of >10 nm, calculated for the three different laser sources, are much broader than what is typically used in such systems. I am not sure if these values are realistic. Given that 1 nm bandwidth corresponds to a minimum pulse energy of 88 µJ (line 195), the fiber laser parameters (PRF: 40 kHz, average power: 10 W hence pulse energy: 250 µJ) suggest a maximum bandwidth of 2.3 nm. For the hybrid fiber laser, the maximum bandwidth is then 6.8 nm. Please check the values given in the text.**

Answer: After verification, there were few mistakes with the numerical applications

- Section 2.3.3, last paragraph, the minimum energy for a 1 nm filter at 300m from lidar and 10 km of altitude is $E_{pmin} = 298$ µJ (This is why on the right figure 3 the fiber laser is on the magenta zone). The difference with the referee value comes from the fact that we forget to write R the background radiance in B, this has been corrected
- Section 2.3.5, the maximum bandwidth for the fiber laser is 0.84 nm and for the hybrid fiber laser it is 2.5 nm. So for the fiber laser, the measure will be slightly affected by the solar background at 300m.

Correction in section 2.3.5 : "… We estimated that the maximum average power of 10 W can be achieved with current technology, it allows the use of a solar filter up to  0.84 nm. Then

the projected wind speed measurement at 300 m will be slightly affected by background noise. The results…"

"… We estimated that the maximum average power of 30 W can be reached, which allows the use of a solar filter up to  2.5 nm. This results…"

Modification line 246 : "In this scenario, a wide bandwidth solar filter can be employed (typically » 1 nm is chosen) "

5. **Line 200: What is the influence of the particle backscattering on the QMZ interferometer output signals? I am wondering if the accuracy of the wind speed retrieval suffers from Mie contamination which is significant at lower altitudes.**

Answer: In presence of particle backscattering, the Speckle noise at the outputs of the QMZ will be more important if no precautions are taken (like mode scrambling in the fiber for example). I complete the appendix A2 to include the influence of the Mie scattering on the retrieved wind speed standard deviation, in response to the commentary of the first referee. However, for my simulation, I problably took a too low value for the particle backscattering, A more realistic value would be between $1.10^{-6}$ and $1.10^{-5}$ (VAUGHAN, J. 1995).

I will redo my simulation with a value of 8.10-6, the median of particle backscattering coefficient at 355 nm ((Herbst, 2016) with the value scaled from (VAUGHAN, J. 1995))

Section 2.3.4, line 211: « … At low altitudes (less than 1 km), we assumed a backscatter coefficient for particles of 8.10^{-6} m^{-1}.sr^{-1}. The simulations… »

Line 214 : "…This error corresponds to  0.12 m/s on the projected wind speed standard deviation. The magenta…"

Section 2.3.5, line 235 : "... This configuration yields a standard deviation sigma_{lidar} of 0.07 m/s at low altitude and 0.17 m/s at high altitude. …"

Section 2.3.5, line 241 : "... The results for standard deviation sigma_{lidar} are 0.05 m/s at low altitude and 0.16 m/s at high altitude."

Section 2.3.5, line 245 : "... This results in a standard deviation sigma_{lidar} of 0.03 m/s at low altitude and 0.09 m/s at high altitude."

6. **Fig. 4: I suggest to mark the two scanning angles (15°, 51°) that are discussed in the text to better visualize the improvement in the RMSE.**

Answer: We agree with the referee and added on the figure on the left (high altitude) 1.36 m/s for 15° and 0.76 m/s for 51 °, and for the figure on the right (low altitude) 2.33 m/s for 15° and 1.22 m/s for 51°.

7. **Perhaps one could also add a 2D color plot, similar to those in Fig. 3, which depicts the RMSE vs. angle and altitude. This would illustrate the influence of scanning angle and altitude on the wind error in a more general manner with the two plots shown in Fig. 3a) and b) representing two intersection curves.**

Answer: It would indeed be interesting to plot this data, as the article does not deal with the influence of altitude on the optimum sweep angle. This is an issue that is currently under study, as we don't yet have a good model for the evolution of Von Karman turbulence parameters as a function of altitude. At present, we use the value defined by aeronautical standards. If we plot this with the current model, there will be no change compared to the figure 5 since the error induced by turbulence prevails.

8. **Can the authors please check the numbers in the parentheses in lines 295f.? I calculate a different value of 2r for θ = 15° when using the equation given in line 279: 2r = 54 m instead of 36 m.**

Answer: Yes it's a typo, I also find 54m after checking.

9. **Line 300: I think it would be helpful for the reader when referring the statement to the equation that contains the term $1/((2\sin(\theta))^2$. I suppose Eq. (3) is meant here.**

Answer: We agree with the referee and added

Line 300 : "… which can be amplified for small scanning angles due to the factor $1/(2\sin(\theta))^2$ in Eq. (3) when retrieving the …"

10. **Line 328: Please mention the respective RMSE values obtained for the two angles in the text to quantify the improvement at the optimized scanning angle.**

Answer: We have changed line 331 to:

Line 331 : "…component recovered with an angle of 15°. The corresponding RMSE obtained for this run of the simulation are 12.9 m/s for an angle of 15° and 7.4 m/s for an angle of 50°. This illustrates …"

11. **Although the example shown in Fig. 5 is illustrative, I am missing information about its representativity for different turbulence scenarios. How does the RMSE at the two different angles vary for multiple runs of the simulation? How does it scale with the variance of turbulence $\sigma^2$?**

Answer:

Ancienne Answer: We do not study the variation of the RMSE with multiple runs of the simulation. This will be done in future work.

Concerning the evolution of RMSE with the variance of turbulence, it becomes rapidly proportional to the turbulence variance since the contribution of lidar measurement error becomes negligible and the different correlation and structure function are proportional to the turbulence variance.

12. **The three terms "scan angle", "scanning angle" and "lidar angle" are used synonymously in the text and should be harmonized to avoid confusion.**

Answer: corrected

**Technical corrections:**

1. **Lines 31, 88: The acronym GLA was already introduced in line 21 and can thus be used here.**

Answer: Corrected line 31 "… and the GLA system …"

Line 88 "… an aircraft for GLA applications,…"

2. **Line 109: Word missing: "… and to focus it into a multimode fiber".**

Answer: corrected Line 109 "… and to focus it …"

3. **Line 119: The acronym OPD was introduced before.**

Answer: corrected line 119 "… increases the OPD for horizontal…"

4. **Line 124: The symbol D_0 should be clarified.**

Answer: we added in line 124: "…, respectively, where D_0 is the OPD for output I_1"

5. **Line 139: Change (Liméry,2008) to Liméry (2018).**

Answer: It has been corrected

6. **Line 188: Correct to "This leads to:".**

Answer: It has been corrected

7. **Line 190: The symbol γ is not described in the text. Also, please check the equation for correctness.**

Answer: line 175 :"… resulting in an overlap function γ equal to 1 across…"

The equation has been check

8. **Fig. 3: The label of the right y-axis should read "Standard deviation on wind speed (m/s)".**

Answer: It has been corrected

9. **Fig. 3: The PRF in the label "Meron C" should be changed from 40 kHz to 400 Hz.**

Answer: It has been corrected

10. **Lines 237, 240: The commas have to be replaced by dots.**

Answer: It has been corrected

**11. Line 239: Correct to "can be reached".**

Answer: It has been corrected

**12. Caption of Fig. 4 can be shortened by writing "[…] for wind measurement at (a) high and (b) low altitude".**

Answer: It has been corrected

**13. Lines 255ff.: The formatting of the symbols in the text should be corrected, e.g. d, r and z should be printed in italics. Conversely, Eqs. (2)-(4), the terms "sin", "cos" and "tan" should be printed upright. This comment also applies to the Appendix sections where upright letters and italics are not used consistently.**

Answer: I hope everything has been corrected correctly.

**14. Line 275: Replace "that is homogeneous and isotropic" with "which is homogeneous and isotropic".**

Answer: It has been corrected

**15. Line 317: The symbol L_0 should be introduced.**

Answer: We replaced L_0  by l in line 320

**16. Line 325: Remove ")" after "Figure 5".**

Answer: It has been corrected

**17. Fig. 5a) Either remove the tick labels from the x-axis or add an axis description (x position in m?). Clarify "plan xOz" in the caption.**

Answer: It has been corrected

*Figure 6: a) Sample of a wind simulation (norm of the wind vector, in the plan xOz where y = 0) and representation of the lidar measurements for the wind reconstruction ahead of a plane b) Evolution of the vertical wind component as a function of the position in the simulated wind volume. In black the actual wind component on the flight path, in red the wind component retrieved with an angle of 15°, in green with an angle of 50 °*

Référence:

HERBST, Jonas. *Development and test of a UV lidar receiver for the measurement of wind velocities aiming at the near-range characterization of wake vortices and gusts in clear air*. 2019. PhD thesis. lmu.

VAUGHAN, J. M., BROWN, D. W., NASH, C., *et al.* Atlantic atmospheric aerosol studies: 2. Compendium of airborne backscatter measurements at 10.6 μm. *Journal of Geophysical Research: Atmospheres*, 1995, vol. 100, no D1, p. 1043-1065.

---

## Author Comment (AC3)

**General response:** We would like to thank referee 3 for his interest in this article, and for the detailed corrections that he suggests to improve the paper's impact on the scientific community. We appreciate that referee 3 says "The manuscript is interesting approach to optimize performance on the lateral and vertical wind measurements of the UV airborne DWL". All his comments have been addressed in the following and the paper have been modified in this regard. We have also added small additional modifications throughout the paper to improve its quality. Please consider this revised manuscript for publication in AMT.

**General comment**

The authors assumed to be the measurement time of 0.1 sec, which corresponds to a range resolution of 25m for an aircraft speed of 250 m/sec. The target measurement range is 100 to 150 m. First concerning is the spatiotemporal and accuracy requirements of the wind measurement for the GLA. What is the response speed meeting the requirements? Please add explanation and references for the requirements.

Answer: The people within the project who specify the requirements have told us that the measurement should be made at a frequency of 10 Hz, about 100 m in front of the aircraft, with an accuracy of 1 m/s. We agree that this needs to be refined with full GLA simulations that take into account the true function of the lidar (and not simplify the lidar model) and study the effect of the lidar's spatial and temporal resolution in order to refine the requirements.

As far as control system response time is concerned, details specific to the GLA are beyond the scope of this article. What we do know is that, at present, the control system and actuators of current GLA systems do not have the time to react fully to turbulence. With lidar measurements at around 100 m and a repetition rate of 10 Hz, the time saving should be sufficient to significantly reduce the turbulence-induced load (see (Fournier et al., 2021)).

Second concern is the signal to noise ratio (SNR) at the focusing region at the lidar angles θ of 10 and 50 degrees. The backscatter coefficient at an altitude of 10 km is not shown in the manuscript. The size for the laser beam and receiver filed at the focusing plane is not shown. What is the SNR and the detectability required for the GLA?

Answer: The molecular backscattere coefficient hasbeen added in the article (see question no15). The size of the laser beam is precise line 181 of the revised version. For the receiver field, it is taken into account in the overlap function. Indeed, when this function is equal to one, the laser beam and the receiver field overlap.

For the GLA, the component of the 3D wind must be estimated with a precision of less than 1 m/s (3 sigma error), so a standard deviation of 0.3 m/s on the component. If we take the case the vertical wind component Vz, estimated with the measurement geometry of four axis as presented in the article, at 100 m in front of the aircraft, this lead to the condition of standard deviation on the projected wind speed $\sigma_{lidar}$< 0.12 m/s for an angle of 15° (see section 2.3.4) and $\sigma_{lidar}$< 0.35 m/s for 50° (present in the revised version). If know we assume that the different lidar component have been designed so that the wind speed measurement is shot noise limited for the considered range of estimation on the lidar axis, we have

$$\sigma_{lidar} = \frac{c\lambda_0}{4\pi D_0} \frac{\sqrt{2}}{SNR * M} \sqrt{1 - \frac{M^2}{4}}$$

If we named $\sigma_{max}$ the maximum value of the std on projected wind speed, the condition is $\sigma_{lidar} < \sigma_{max}$ and become on the SNR:

$$\frac{c\lambda_0}{4\pi D_0} \frac{\sqrt{2}}{\sigma_{max} * M} \sqrt{1 - \frac{M^2}{4}} < SNR$$

$D_0 = 3\ cm$, $M$ the contrast of interferences due to the width of the Rayleigh spectrum is equal to 0.68 at 10 km of altitude. So for $\sigma_{max}$ = 0.12 m/s, the minimum SNR is 4600 and for $\sigma_{max}$ = 0.35 m/s, the minimum SNR is 1580.

This is achieved with the three laser designs established in the article.

Regarding to the second concern, third concern is repetitiveness of the wind filed. The θ of 10 and 50 degrees have different volumes. When the laser beam and the receiver filed are focused, the reviewer thinks that the wind filed at the focusing volume will be no longer the repetitiveness around the plane. The refractive turbulence is strong in the UV.

Answer: Concerning the repetitiveness, it is taken into account in the simulation with the movement of the plane during the measurement time. This induces a displacement of the lidar axis along flight direction and the wind is integrated.

We are sorry but we are not sure to understand the sentence "When the laser beam and the receiver filed are focused, the reviewer thinks that the wind filed at the focusing volume will be no longer the repetitiveness around the plane"

For the effect of the refraction induce by turbulence, the effect should be minor since at short distance (100m -300 m), the correlation of the turbulence is good.

**Specific Comments**

1. Line 17: Introduction. Please add references regarding to aviation accident and safety.

Answer:  We are not sure which part of the introduction the referee mentions, but we assume it relates to the sentence in line 30. We propose the following reference line 31 (see revised version) "In addition, it will limit aircraft vibrations, particularly the effects of air pockets that can30 hurt passengers (Kaplan et al., 2005)."

2. Lines 26-27: "100m- 200m ahead of the aircraft" is the spatial requirement. Please add explanation and reference related to spatial requirement.

Modification line 37: "100 m - 200 m ahead of the aircraft (In the case of the Airbus XRF1 (Fournier et al., 2021), the optimal distance ahead of the aircraft is 91 m, giving the control system enough time to react) "

3. Lines 31 and 88: "abbreviation for Gust Load Alleviation" is shown again. Please remove "Gust Load Alleviation" if you use the abbreviation GLA.

Answer: it was corrected

4. Line 55: QMZ should be "Quadri Mach-Zehnder (QMZ)".

Answer: it was corrected

5. Lines 62-63: "a factor √2 increase in statistical error" is not clear. Please add explanation regarding on the factor and the reference.

Answer: The factor appears when the number of channel of higher than 2 (for QMZ and fringe imaging technics). For the QMZ, It can be seen by the fact that 2 channel are required to determine the wind speed, those in phase opposition and localize near the maximum sensitivity of interference intensity with Doppler shift (see figure 3 in the revised version). So half of the collected photons are use to determine the wind speed, hence the √2 factor on SNR and so on the wind speed standard deviation

Modification line 62: "(at the cost of a factor √2 increase in statistical error due to the desensitization of the interferometer to the backscattering ratio (Bruneau, 2001, 2002), and do not require laser stabilization)"

6. Line 110: How much is the numerical aperture (NA) of multimode fiber assumed in the manuscript?

Answer: The NA of the multimode fiber is 0.22 (half aperture angle 12.7 ° ), that match with the telescope aperture of f/D=4 (half aperture angle 7.12° < 12.7)

Modification line 112: "…and to focus it into a multimode fiber with a numerical aperture of  0.22."

7. Line 156: What is "moy"?

Answer: It is "moyenne" for "average" in French, it was corrected by "av"

8. Line 162: "Maximum Likelihood Estimator (MLE)". Please add reference.

Answer: modification line 171 :" The Maximum Likelihood Estimator (MLE) (see paper of Cézard et al. (2009) for principle) was then employed…"

9. Line 165: "Cramer Rao's lower bound". Please add reference.

Answer: modification line 171 :" … the analytical formula closely matched Cramer Rao's lower bound (Cézard et al., 2009) as we obtain…"

10. Lines 168-169: The laser beam shown in Figure 2(a) should be convergent.

Answer: It was corrected

11. Lines 170-171: Please add explanation regarding on relation between F-number and the NA.

Answer: The noise factor take into account the noise added during the electronic amplification process (fluctuation of the number of electrons in the cascade produce in PMT and APD). According to Hamamatsu in his "PMT_HandBooK V4", the noise figure is define by the square of the ratio of SNR at the input to the SNR at the output. In our case, taking the photon noise on the detector, we have $SNR_{input} = \frac{N}{\sigma_i}$ with N the number of photon and $\sigma_i = \sqrt{N}$. At the input of the detector after the amplification, we have $SNR_{output} = \frac{G\eta N}{\sigma_o}$ with $\eta$ the quantum efficiency and G the gain. So the noise factor will be $F = \left(\frac{SNR_{input}}{SNR_{output}}\right)^2 = \left(\frac{\sigma_o}{G\eta\sigma_i}\right)^2$. So the noise variance at the output of the detector produce by photon noise is $\sigma_o{}^2 = F(G\eta\sigma_i)^2 = F(G\eta)^2 N$. The square over $\eta$ has been forgot in the first version

Modification line 189: "Taking the definition of the excess noise factor from (PMT Handbook), we have $F = \left(\frac{SNR_{input}}{SNR_{output}}\right)^2$ with $SNR_{input} = \frac{N}{\sigma_i}$ and $SNR_{output} = \frac{G\eta N}{\sigma_o}$. N represents the sum of backscattered photons obtained on the four detectors, G stands for the gain of the detector, and η

signifies the quantum efficiency. The noise variance induced by the backscattered shot noise at the output of the detectors will be is $\sigma_o^2 = F(G\eta\sigma_i)^2 = F(G\eta)^2 N$. This noise must exceed the detection noise, leading to the condition $N \gg \frac{4\sigma_{det}2}{F(G\eta)2}$, where σ2 det denotes the detection noise of a detector expressed in the number of electrons calculated for a range gate of 25 m. In order to meet the condition, we take $N > 10\frac{4\sigma_{det}2}{F(G\eta)2}$, with the right term corresponding to the equivalent number of photons produced by the detection noise. For the PIN, we found 1.1 × 108, for the APD 2.8 × 106 and for the PMT 2.9 × 10−5. Only the PMT ensures a low level of detection noise compared to the shot noise le"

12. Line 172: I don't understand M2<8. Please add explanation regarding on physical meanings of M2 and M2<8.

Answer: The $M^2$ (or M squared) is a parameter that define the laser beam quality. If we take the ideal case of a Gaussian beam, the half beam divergence is $\theta_0 = \arctan(\frac{\lambda}{\pi\omega_0})$ with $\lambda$ the wavelength and $\omega_0$ the beam waist. In practice, it is hard to reach the perfect Gaussian beam, that is traduced by an increase of the beam divergence relative to $\theta_0$. The $M^2$ is then used to quantify the increased of the divergence and is $M^2 = \theta/\theta_0$.

Modification line 180: "… where $M^2$, define as the ratio of the beam divergence angle to the beam divergence angle of the perfect Gaussian beam at the same wavelength, is considered lower than 8, value obtain for the commercial laser Merion C by Lumibird."

13. Line 190: What is "γ(r)2"? Please add explanation. Is "γ(r)2" correct?

Answer: it is the overlap function, the definition was added line 147

14. Lines 194, 236, and 240: "0," should be "0.".

Answer: it was corrected

15. Lines 199-200: The backscatter coefficient at an altitude of 10 km is not shown in the manuscript. Please add the backscatter coefficient.

Answer: modification line 217: "The backscatter coefficient for molecules is 7.2 × 10−6 m−1 sr−1 on the ground and 2.1 × 10−6 m−1 sr−1 at 10 km of altitude."

16. Line 200: "m-1.sr-1" should be "/m/sr".

Answer: I made a mistake in the format in which I wrote the units, AMT indicates that the unit in the denominator must be formatted with negative exponents. I correct it for all values with units

17. Line 203: Laser has a spectral width of <500 MHz. How long is the pulse width assumed? Is it possible to develop the laser system?

Answer: The pulse width is assumed to be 10 ns, I also made a mistake in the value, the laser have a full width at 1/e$^2$ of 400 MHz. The value taken here for laser parameters are the main one to realize the simulations. We prefer not to go into detail about the laser, as this is not the purpose of this article.

18. Line 204: Please add references regarding the spectral broadening of 3 GHz.

Answer: modification line 219 "… by the thermal movement of the molecules (6.3 GHz for a full width at 1/e2 (Bruneau and Pelon, 2003))"

19. Line 207: Figure 3 shows result of the relation between laser average power and pulse repetition frequency (Hz). Do you need to show the results at the laser average power of >10W operating at PRF of < 100 Hz. Is it feasible to develop the laser system in the term of the laser power density and laser-induced optical damage?

Answer: We agree that some of the laser parameters may lead to laser design that are not feasible in practice, for technological considerations as the one you mention. Maybe the range the laser parameters could be refined taking into account its technical considerations, but it is not the goal of the article to described all this, that could made the subject of other article.

20. Line 230: "repetition rate of 400 Hz and delivers 22.5 mJ of energy per pulse". In Figure 3, Merion C is "9W 40 kHz". Which is it correct?

Answer: It was a typos, corrected

21. Lines 225-240: Do three lasers have enough tolerance for the optical damage? Does each laser have enough tolerance for the optical damage?

Answer: For the Merion C, the optics of the emission telescope have been chosen so as not to be damaged by the high energy of each pulse emitted by the laser.

For the fiber laser, these designs are currently studied by laser team at ONERA, more details on this technology will be available soon

22. Figure 4: Please embed "high altitude" and "low altitude" int the Figures 4(a) and 4(b), respectively.

Answer: It was corrected

23. Line 270: "root mean square error". "root mean square error (RMS)" should be better.

Answer: It was corrected

24. Line 297: "root mean square error" should be removed.

Answer: It was corrected

25. Line 325: "Figure 5) displays the results.". ")" should be deleted. "results", how did you simulate? It is not clear. Please add explanation. Did you investigate the statistic difference between real wind component and retrieved wind component at two angles of 15 and 50 degree? Please add results and explanation.

Answer: It was corrected.

We indeed forgot to mention the parameters used to obtain the results. We propose the following modification line 342 : "For the simulation, the wind box was taken equal to 8 km×800 m×800 m, sample every 5 m in each directions.The turbulence length scale is taken equal to 762 m, value at 10 km, and the turbulence is 100 m2 s−2. We considered that the plane is moving at 250 m s−1, centered at y = 0 and z = 0, along x axis, and that the lidar is located in the nose of the aircraft. The model use for the lidar is simplify, considering only one measurement of the projected wind speed at a range z = d/ cos (θ), over a range gate of 25 m. The model of the lidar measurement noise is assumed to be Gaussian with the projected wind speed obtain on the range gate for the mean and a standard deviation corresponding to $\sigma_{lidar}$. For the simulation we take the one obtain for the Merion C. Figure 6 … "

The statistics difference between real wind component and retrieved wind component at two angles of 15 and 50 degree has not yet been studied in detail. Due to short time to answer the comments (I made mistakes in reading the comments and the comments of referee no3 has been discovered lately) we will not be able to perform simulations and give more results.

**Refences**

26. Line 445. What is the title? Which journal is the manuscript reviewed?

Answer: The title and journal have not yet been determined, as the document is currently under development.

27. Line 459. 2022 -> 2021.

Answer: It was corrected

**Miscellaneous**

28. The summary of specification parameters used in the simulations will be helpful for the readers. Please add the summary of the specification parameters.

Answer: Table that summurarize all parameters have been added in the appendix.

---

## Author Response (AR2)

editor comments :

Dear authors,

Thank you very much for the revision which has been well received by both original reviewers. One of the reviewers has three remaining points which I would like you to address. Also, the other reviewer noted that he/she had a typo in the original review and sent me the following note:

"In my last comments, I commented as follows: When the laser beam and the receiver filed are focused, the reviewer thinks that the wind filed at the focusing volume will be no longer the repetitiveness around the plane. The refractive turbulence is strong in the UV.
"the repetitiveness" was typo. "representativeness" is correct. If possible, I recommend to add an explanation sentence in the body."

I would thus welcome if you addressed also this comment in your final revision which I will review myself and then take a final decision on your manuscript.

Thanks in advance and best regards,

Markus Rapp
Assoc. Editor

General answer; We would like to thanks the editor and the reviewers for the revisions that will improve the article. We appreciate that "the revision has been well received by both original reviewers". We have addressed all of the concerns in the revised manuscript and have detailed them in the report below. Please consider this revised manuscript for publication.

**Reviewer: "In my last comments, I commented as follows: When the laser beam and the receiver filed are focused, the reviewer thinks that the wind filed at the focusing volume will be no longer the repetitiveness around the plane. The refractive turbulence is strong in the UV.**
**"the repetitiveness" was typo. "representativeness" is correct. If possible, I recommend to add an explanation sentence in the body."**

Answer:

We agree with the reviewer that the turbulence can affect the propagation of the laser beam in the atmosphere. This can result in beam wandering and difference between the expect focusing position and the real one. However, if this motion is smaller than 25 m, we don't think that this will affect significantly the results. Indeed, we have already considered the fact that, for each measurement integrated over 0.1 s, the focusing position changes over 25 m due to the plane motion (at 250 m/s). In the simulation, this had little effect on the error of the reconstructed wind velocity. This is likely due to the fact that, for a "Von Karman" statistical model, at high frequency, the amplitude of the harmonic get very small (so the wind does not vary significantly over short distances). However, this effect will be addressed in more detail in future studies that will account for beam refraction in turbulences.

To support the explanations, and to be consistent with the simulations described after in the paper, we remade the calculations of the RMSE as a function of lidar angle for a Von Karman turbulence with $\sigma = 10\ m/s$. This implies the following modifications:

- The figure 5 was updated with a standard deviation of the wind amplitude of $\sigma = 10\ m/s$
- We modify line 315 as follow: "… a Von Karman turbulence with l equal to 762 m (2500 ft) and $\sigma_S$, the standard deviation of the wind amplitude equal to 10 m s-1 …"
- We modify line 317 as follow: "The RMSE obtained for this angle is 7.2 m s−1, nearly twice as low as the RMSE of 12.7 m s−1 obtained for an angle of 15°"
- We modify line 330 as follow: "Additionally, we assume a turbulence strength $\sigma_S$ of 10 m s-1"
- We modify line 350 as follow : " … we assume a turbulence with a standard deviation of the wind amplitude of 10 m s-1 "
- We propose to add line 351: "…located in the nose of the aircraft. In particular, for each measurement, we account for the plane motion during the integration time, that is to say the slight variation of the measured projected wind observed by each laser pulse…."
- We propose to add line 366: "… . This illustrates the improvement achieved with the optimized lidar angle of 50◦. In addition, the result is close to the theoretical RMSE given in Fig. 5b). This shows that the motion of the plane during the integration time (i.e. 25 m) has little effect on the error in the wind reconstruction."
- And Line 402 in the conclusion : "The effect of refraction due to turbulence has not been taken into account in the 3D wind simulator. In the UV, the refraction is strong and the beam can be significantly deflected as it propagates through the atmosphere. This can lead to an increase in the size of the probe volume, depending on the direction of the refraction. This will be addressed in future studies."

The authors have adequately addressed most of the issues I raised and revised the manuscript accordingly. I recommend publication after minor revisions as follows:

1. The major limitations of the end-to-end simulator, discussed in the response letter, should be included in the manuscript. Specifically, the assumption of a perfect interferometer with a contrast of 1, and the fact that reduced contrast due to optical imperfections will be addressed in a refined simulator, are noteworthy and should be mentioned.

2. In Fig. 5, the RMSE values should use the correct punctuation, for example, 2.33 m/s instead of 2,33 m/s. Additionally, units should be written in exponential form (m s⁻¹) in the axis labels to comply with AMT style guidelines. This also applies to some labels in other figures.

3. It is unfortunate that the authors did not study the reproducibility or variability of the RMSE across multiple simulation runs. They should at least provide an estimate of how the RMSE values vary and indicate that a more detailed sensitivity study will be performed in the future.

1. **Referee: The major limitations of the end-to-end simulator, discussed in the response letter, should be included in the manuscript. Specifically, the assumption of a perfect interferometer with a contrast of 1, and the fact that reduced contrast due to optical imperfections will be addressed in a refined simulator, are noteworthy and should be mentioned.**

Answer: We add in the conclusion line 381: "…to refine the calculation of the performances of the system. Currently we assume a perfect interferometer with all transmission of the optics equal to 1 and an instrumental contrast of 1. This will be reevaluated in future studies with transmission and contract measured experimentally."

2. **Referee: In Fig. 5, the RMSE values should use the correct punctuation, for example, 2.33 m/s instead of 2,33 m/s. Additionally, units should be written in exponential form (m s⁻¹) in the axis labels to comply with AMT style guidelines. This also applies to some labels in other figures.**

Answer: it has been corrected

3. **Referee: It is unfortunate that the authors did not study the reproducibility or variability of the RMSE across multiple simulation runs. They should at least provide an estimate of how the RMSE values vary and indicate that a more detailed sensitivity study will be performed in the future.**

Answer:

In this study, the RMSE is actually evaluated by comparing the reconstructed wind with the actual wind along the flight path, that is to say, over ~300 independent cases (that represent

approximately 8 km divided by 25 m the size of the volume integrated over 0.1s at 250 m.s$^{-1}$) at each run of the simulator.

We followed the advice of the referee, whom we thank, and. perform multiple run of simulation to have a better evaluation of the RMSE. Results of RMSE obtained with 180 runs of the simulations show a mean value of 12.7 m.s$^{-1}$ with a $3\sigma$ error of 0.3 m.s$^{-1}$ for the angle of 15° and a mean value of 7.2 m.s$^{-1}$ with a $3\sigma$ error of 0.15 m.s$^{-1}$ for the angle of 50°.

To explain this point, we modified line 358: "…for the Merion C. Once the wind components are estimated, the RMSE is estimated at each simulator run, using the mean value of the squared differences between the estimated wind component and the true wind component over the flight path over 312 values (~8 km/25 m), and taking the square root of the resulting value. Figure 6 displays the results…."

We remove the sentence line 364 and add the following explanation: ". 180 simulation runs have been performed and statistics on all obtained RMSE shows a mean value of 12.7 m.s$^{-1}$ with a $3\sigma$ error of 0.3 m.s$^{-1}$ for the angle of 15° and a mean value of 7.2 m.s$^{-1}$ with a $3\sigma$ error of 0.15 m.s$^{-1}$ for the angle of 50°. This illustrates the improvement…"

Additional modifications to improve the readability of the article.

- $\sigma^2$, define as the variance of the turbulence, can be confusing and not well defined in the text. Moreover, $\sigma$ appears to many times in the article, for different physics parameters and mathematics function. So we decided to explained it as $\sigma_S$ the standard deviation of the wind amplitude in the turbulence.

- We modify line 310 as follow: " … $\sigma_S$ the standard deviation of the wind amplitude in the turbulence, l the turbulence length scale,…"

- We add line 311 the following explanation on $\sigma_S$: "The standard deviation of the wind amplitude is related to the spectrum energy E_v(k) of the wind field with the equation $\int_0^\infty E_v(k)\,dk = \frac{3\sigma_S}{2}$ (Wilson,1998), where k represents the spatial frequency."

- To avoid repetition, we delete line 340 the following sentences: "… represent the direction (x, y, or z), , and δij is the Kronecker delta…"

- In equation (6), (7), (8) and all other equations where this physics quantity appears, $\sigma$ has been replaced by $\sigma_S$.

- We modify line 340 as follow: "For Von Karman turbulence, Ev (k) = $1.4528\frac{\sigma_S{}^2 k^4 l^2}{(1+k^2 l^2)^{17/6}}$ . (already define line 315)"

- We correct line 350 : ".. 762 m, value at 10 km of altitude, and .."

- We modify line 352 as follow: "…Vaircraft =250 m s−1, that it is centered at y = 0 and z = 0, along x axis, and that the lidar is located in the nose of the aircraft…"

- We modify line 355 as follow: "We used a simplified model for the lidar, considering, at each pulse, only one measurement on the laser axis of the projected wind speed at a range z = d/ cos (θ) over a range gate of 25 m."

- We correct line 362 : "… In Fig. 6.b), the green line, that corresponds to the vertical component retrieved using a lidar angle of 50…"

- We correct line 362 :"… using a lidar angle of 50∘, is closer to the black line, that represents the real wind, than the red line, that corresponds to the vertical component reconstructed using a lidar angle of 15∘ …"